# Zooming without Zooming: Region-to-Image Distillation for Fine-Grained Multimodal Perception

Lai Wei [1 2 3 *]  Liangbo He [2 *]  Jun Lan [2 ‡]  Lingzhong Dong [1 2]  Yutong Cai [1]  Siyuan Li [2]  Huijia Zhu [2]
Weiqiang Wang [2]  Linghe Kong [1]  Yue Wang [3]  Zhuosheng Zhang [1]  Weiran Huang [1 4 †]

## Abstract

Multimodal Large Language Models (MLLMs) excel at broad visual understanding but still struggle with fine-grained perception, where decisive evidence is small and easily overwhelmed by global context. Recent "Thinking-with-Images" methods alleviate this by iteratively zooming in and out regions of interest during inference, but incur high latency due to repeated tool calls and visual re-encoding. To address this, we propose *Region-to-Image Distillation*, which transforms zooming from an inference-time tool into a training-time primitive, thereby *internalizing* the benefits of agentic zooming into a single forward pass of an MLLM. In particular, we first zoom in to micro-cropped regions to let strong teacher models generate high-quality VQA data, and then distill this region-grounded supervision back to the full image. After training on such data, the smaller student model improves "single-glance" fine-grained perception without tool use. To rigorously evaluate this capability, we further present *ZoomBench*, a hybrid-annotated benchmark spanning six fine-grained perceptual dimensions, together with a dual-view protocol that quantifies the global-regional "zooming gap". Experiments show that our models achieve leading performance across multiple fine-grained perception benchmarks (Figure 1), and also improve general multimodal cognition on benchmarks such as visual reasoning and GUI agents. Our code is available at https://github.com/inclusionAI/Zooming-without-Zooming.

*Equal contribution ‡Project lead [1]School of Computer Science, Shanghai Jiao Tong University [2]Ant Group [3]Zhongguancun Academy [4]Shanghai Innovation Institute. Correspondence to: Weiran Huang <weiran.huang@sjtu.edu.cn>.

*Proceedings of the 43rd International Conference on Machine Learning*, Seoul, South Korea. PMLR 306, 2026. Copyright 2026 by the author(s).

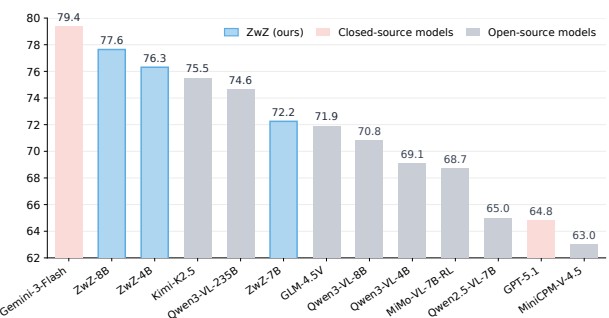

*Figure 1.* Average scores across multimodal perception benchmarks. ZwZ-4B/7B/8B demonstrate competitive performance compared with current SOTA MLLMs (e.g., Gemini-3-Flash, Kimi-K2.5, Qwen3-VL-235B).

## 1. Introduction

Multimodal large language models (MLLMs) achieve strong general visual understanding and reasoning (Bai et al., 2025a; Comanici et al., 2025), yet they remain brittle on fine-grained perception tasks where the decisive evidence is small and easily overwhelmed by the global context (Wang et al., 2025c), such as tiny text and subtle attributes. In a standard single-pass setting, models must retrieve such micro evidence from thousands of visual tokens, making it difficult to reliably isolate minute evidence amid clutter, distractors, and redundant information.

To mitigate this, recently popular paradigms like *"Thinking-with-Images"* (Wu et al., 2025a; Wu & Xie, 2024; Zhang et al., 2025h; Zheng et al., 2025b) equip MLLMs with agentic tool use to iteratively *zoom in and out* regions of interest and re-encode them during inference. Its effectiveness largely comes from reducing interference: once a micro-crop is isolated, the model can devote its capacity to a small region, turning a needle-in-a-haystack problem into straightforward recognition. However, this comes with substantial latency due to repeated tool calls and multiple visual encoding passes, limiting real-time usage. This motivates us a natural question: *can we obtain the accuracy benefits of zooming in while keeping inference to a single forward pass?*

We achieve this by introducing a novel and easy-to-implement *Region-to-Image Distillation* method. Concretely, we decouple "zooming in" from inference-time (i.e., the agentic "Thinking-with-Images" process) and repurpose it as a training-time primitive. Firstly, we employ advanced teacher models (large, cloud-hosted) to generate perception-centric questions and high-consensus answers only on micro-crops, where fine details are unambiguous and hallucinations are minimized. Then, these region-grounded data are *distilled* back to the full image, with explicit visual grounding via bounding-box overlays to avoid referential ambiguity. Training on such distilled supervision teaches the student model (smaller, localized) to suppress global noise and pay attention to micro-level evidence directly from the full image. Therefore, the student model internalizes the benefits of agentic "zooming in" into a single *glance* (one forward pass on the full image) at inference time without iterative tool use. This can also be summarized as an idea of *"Zooming without Zooming"* shown in Figure 2. Here, the first "Zooming" refers to the training-time primitive: we zoom into micro-regions to synthesize fine-grained training data and thus our model can "zoom in and out" in mind. In contrast, the second "Zooming" denotes the inference-time tool-use we seek to bypass.

To pave the way for efficiently building a high-quality benchmark at low cost, as well as rigorously evaluate fine-grained acuity in this regime, we present *ZoomBench*. It is a challenging benchmark of 845 high-quality VQA samples built with our synthesis pipeline plus human verification, spanning six perceptual dimensions and covering diverse question formats. Unlike from prior benchmarks, *ZoomBench* naturally supports a dual-view evaluation protocol to quantify the global-regional "zooming gap", along with interpretability analyses based on relative attention maps (Khayatkhoei et al., 2025) for deeper inspection.

We conduct extensive experiments to validate the effectiveness of our region-to-image synthetic data. By leveraging reinforcement learning (Guo et al., 2025) on just 37k data, our models (ZwZ) achieve substantial performance gains across all evaluated fine-grained perception benchmarks, including our proposed *ZoomBench*. Remarkably, ZwZ-8B outperforms similarly sized baselines, and remains competitive with or even exceeds significantly larger state-of-the-art MLLMs (e.g., Kimi-K2.5, Qwen3-VL-235B, and Gemini-3) on many challenging perceptual tasks despite using much smaller backbones (Figure 1). Our models also consistently surpass many agentic "Thinking-with-Images" models such as DeepEyes (Zheng et al., 2025b) and Thyme (Zhang et al., 2025h), while avoiding their iterative tool-use latency (Figure 4). Beyond in-distribution perception, our approach further enhances models' general multimodal cognition, improving out-of-distribution generalization on visual reasoning, AIGC detection, and GUI agent benchmarks. Diving

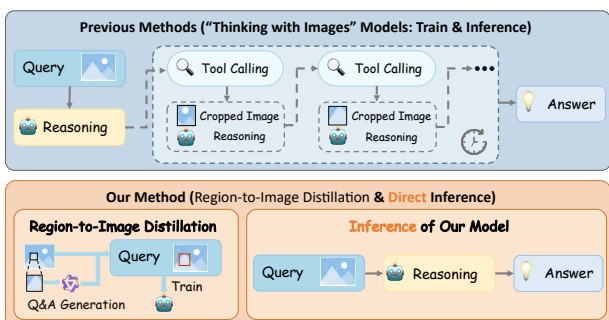

Figure 2. **Zooming without Zooming.** "Thinking-with-Images" models rely on iterative tool-based cropping and re-encoding at inference, incurring high latency. Our *Region-to-Image Distillation* performs zooming only during training to synthesize region-grounded supervision on the full image, enabling single-pass fine-grained perception without test-time tool use.

deeper, our dual-view evaluation and attention map analysis show that ZwZ better localizes task-relevant micro evidence from the full image, effectively narrowing the global–regional "zooming gap". Finally, we discuss the comparison and boundary between ZwZ and "Thinking with Images", clarifying when tool-based image operations are necessary and when their benefits can be distilled into a single forward pass.

To summarize, our key contributions are the following:

- We propose *Region-to-Image Distillation*, which utilizes the expertise of MLLMs on cropped region to synthesize training data conditioned on the full image, thereby distilling the benefits of tool-based "zooming in" into a single *glance* at inference .

- We introduce *ZoomBench*, a diverse and rigorous benchmark for fine-grained multimodal perception, featuring broad coverage across multiple scope dimensions, a human–AI hybrid construction pipeline, and various evaluation protocols.

- Experiments show that our method substantially improves fine-grained perception, exceeding tool-based agentic models while achieving significantly lower inference latency.

**Conflict of Interest Disclosure.** The authors declare no financial conflicts of interest.

## 2. Related Works

**Fine-Grained Multimodal Perception.** Recent advancements in fine-grained multimodal perception have shifted towards a *"Thinking-with-Images"* paradigm, where MLLMs actively acquire local information during inference rather than relying solely on a global image encoding (Hou et al., 2025; Hu et al., 2024; Jiang et al., 2025a; Ma et al., 2026;

Wang et al., 2025g; Wei et al., 2025c;d; Wu et al., 2025b; Yu et al., 2025c; Zhang et al., 2025c;d;f; 2026; Zhou et al., 2024). Notably, DeepEyes (Zheng et al., 2025b) and Mini-o3 (Lai et al., 2025) employ reinforcement learning to incentivize the usage of visual tools such as "Zoom in (Crop)" and "Search", while Thyme (Zhang et al., 2025h) and Pixel-Reasoner (Su et al., 2025) enable models to generate code or perform pixel-space operations to manipulate visual inputs dynamically. Other training-free approaches (Fu et al., 2025; Hu et al., 2024; Khayatkhoei et al., 2025; Liu et al., 2025a; 2024; Luan et al., 2026; Mondal et al., 2024; Peng et al., 2025; Shen et al., 2025) employ tree search strategies or attention mapping to let the model zoom in the key region during inference. Though effective, this paradigm suffers from a significant inference overhead. The requirement for multiple visual encoding passes or iterative tool call loops increases latency, making these methods less suitable for real-time applications. To this end, we propose a *Region-to-Image Distillation* approach which aims to internalize the benefits of visual tools through pure RL training, without the need for explicit tool invocation or recursive processing during inference.

**Multimodal Synthetic Data.** While datasets like Visual Genome (Krishna et al., 2017), VStar (Wu & Xie, 2024), Rexverse (Jiang et al., 2024), Thyme-SFT/RL (Zhang et al., 2025h), VGR-SFT (Wang et al., 2025d), and Visual Probe (Lai et al., 2025) involve valuable fine-grained perception data, their utility is significantly constrained by limited sample sizes, low difficulty for current MLLMs, or homogeneous image, task distributions, largely due to their reliance on manual annotation. This scarcity necessitates automated and scalable synthesis pipelines to derive high-quality VQA data from vast, unlabeled image corpora (Cai et al., 2025; Gao et al., 2025; Lin et al., 2026a;b; Shi et al., 2025; Wei et al., 2025b). Recent methods such as Genixer (Zhao et al., 2024), MM-Evol (Luo et al., 2025), and Oasis (Zhang et al., 2025b) utilize proprietary MLLMs (e.g., GPT-4 (Hurst et al., 2024)) to synthesize large-scale visual instructions. However, they predominantly rely on a "global-to-global" synthesis pipeline based on traditional "teacher-to-student" distillation, where teacher models are prompted with full images to generate QA pairs based on further workflows. This can easily lead to hallucinatory artifacts and poor perceptual grounding, as even advanced MLLMs struggle to discern minute details when facing a coarse-grained image. Unlike these prior methods, our *Region-to-Image Distillation* targets the "zooming gap" in MLLMs by adopting a region-centric synthesis strategy: we generate QA pairs from scratch using micro-crops as anchors to ensure grounded factuality, while training the model on the full image.

**Perception Benchmarks.** The evaluation of fine-grained perception has led to several pioneering benchmarks (Tong

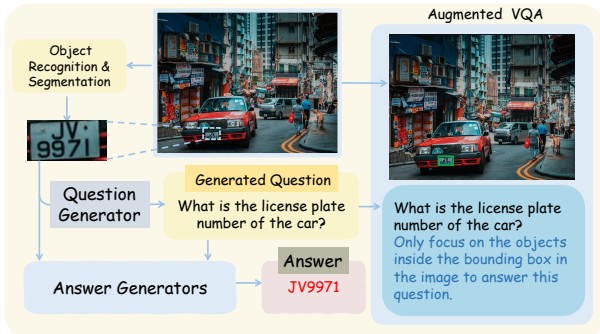

*Figure 3.* Overview of *Region-to-Image Distillation*. We synthesize fine-grained VQA pairs on zoomed-in micro-crops using strong teachers as generators, then distill them to the full image via box-overlay grounding and an augmented prompt, enabling improved single-pass inference without test-time zooming.

et al., 2024; Wei et al., 2024; Yue et al., 2024). However, CV-Bench (Tong et al., 2024), VStar (Wu & Xie, 2024), and HR-Bench (Wang et al., 2025f) are limited by narrow task coverage, single evaluation protocol, or overly templated question styles. CountQA (Tamarapalli et al., 2025), RC-Bench (Niu et al., 2025), ColorBench (Liang et al., 2025), and GroundingME (Li et al., 2025d) only target specific fine-grained capabilities such as counting, Document OCR, color perception, and visual grounding. MME-RealWorld (Zhang et al., 2025g) and TreeBench (Wang et al., 2025b) covers broader sub-tasks, but they highly rely on labor-intensive and consuming manual construction. Our proposed benchmark, *ZoomBench*, addresses these shortcomings by leveraging our proposed *Region-to-Image Distillation* method to reduce human annotation cost. Beyond standard benchmarking, we also include two evaluation protocols for deeper analysis: (1) dual-view evaluation and (2) relative attention map evaluation.

## 3. Zooming without Zooming

To instantiate the "Zooming without Zooming" idea, we make two primary contributions as follows: (1) our *Region-to-Image Distillation* method, which internalizes the benefits of tool-based zooming during training, and (2) the *ZoomBench*, which is designed to rigorously and comprehensively evaluate this capability.

### 3.1. Region-to-Image Distillation

We propose *Region-to-Image Distillation (R2I)* as shown in Figure 3, an automated and scalable synthesis pipeline capable of deriving high-veracity VQA data from vast, unlabeled image corpora. It aims to distill fine-grained, region-grounded supervision into full-image VQA instances, enabling *single-pass* fine-grained perception at test time without any tool use.

**Setup.** In fine-grained perception, the decisive evidence is often confined to a micro-region $R$ in the image $I$, which is small and easily overwhelmed by the global context. Our goal is to optimize the student model by distilling the teacher's regional-view expertise on $R$ into the student's global-view predictions on $I$, which is defined as *Region-to-Image Distillation*. We explain the details of our proposed method as follows.

**(1) Zoom-in Synthesis on the Cropped Region.** Given an image pool $\mathcal{D}_{\text{raw}}$, we define a "zoom-in" function $\mathcal{P}(I)$ that generates a set of candidate bounding boxes $\{B_1, B_2, \ldots, B_n\}$ using an object recognition and segmentation system. For each box $B_i$, we extract the corresponding region $R_i$ by cropping and resizing. Importantly, $\mathcal{P}$ is *object-centric*: it proposes regions, each covering at least one visible object in the image, so these extracted regions are semantically meaningful and more likely to contain informative visual evidence. To ensure the synthetic data specifically target fine-grained challenges, we enforce that: $\mathcal{R}_{\text{target}} = \left\{ R_i \mid \frac{\text{Area}(B_i)}{\text{Area}(I)} < \tau \right\}$, where $\tau$ is a sparsity threshold (e.g., $\tau = 0.1$). This filtering ensures that the evidence is "hidden" in the global view. For each micro-region $R \in \mathcal{R}_{\text{target}}$, a teacher model serves as a question generator to propose perception-centric questions $\mathcal{Q}_R$ that are strictly answerable from $R$ alone. These questions rely on fine-grained cues that are easily overlooked from the global view, such as tiny text, symbols, subtle attributes or small-instance counts. To obtain high-veracity labels without manual effort, we utilize teacher models as answer generators to derive the pseudo-label $A$ via majority voting for each question $Q \in \mathcal{Q}_R$. Note that a triplet $(R, Q, A)$ is retained only if the answer generators reach a high consensus, substantially reducing hallucinated or invalid questions.

**(2) Zoom-out Distillation to the Full Image.** After obtaining reliable region-level supervision, we zoom out to distill these QA pairs into their original full-images, forming fine-grained training instances. However, a question $Q$ that is unambiguous when isolated within $R$ may become referentially ambiguous in the global image $I$ (please see Appendix E.1 for examples). To resolve this referential uncertainty, we apply a grounding transformation $\mathcal{G}(I, Q, B)$ (also can be viewed as $\mathcal{P}^{-1}$, an inverse transformation of $\mathcal{P}$) that yields an augmented training pair $(I', Q')$. Specifically, $\mathcal{G}$ overlays the visual bounding box $B$ onto $I$ and appends a spatial constraint to $Q$. Consequently, the training triplet is formulated as $(I', Q', A)$ (e.g., the "Augmented VQA" in Figure 3), where $I'$ and $Q'$ ensure the question remains precisely anchored to the intended region. Furthermore, we perform difficulty filtering on the synthetic data using a smaller multimodal model, removing overly easy samples and yielding the final training dataset $\mathcal{D}_{\text{syn}}$. Finally, the

student training objective is formulated as:

$$\max_{\theta} \ \mathbb{E}_{(I', Q', A) \sim \mathcal{D}_{\text{syn}}, \widehat{A} \sim \pi_{\theta}(\cdot \mid I', Q')} \left[ r(\widehat{A}, A) \right], \quad (1)$$

where $r(\widehat{A}, A)$ is a task reward and $\pi_{\theta}$ is the student policy.

**Intuitive Explanation.** A natural question then arises: can the model still improve its performance at inference time when the bounding box is removed? To answer this, we can treat $B$ as *privileged information* (Pechyony & Vapnik, 2010; Vapnik & Vashist, 2009) available only during training. This learning paradigm has been both theoretically and empirically verified in prior works (Bartolo et al., 2026; Collier et al., 2022; Jiang et al., 2025b; Sharmanska et al., 2014). Formally, the model learns a conditional distribution $P(A \mid I, Q, B)$ (nearly equivalent to $P(a \mid I', Q')$ as we use $I', Q'$ as an operational way to encode $I, B$) during training. Through visual grounding (box overlays), $B$ acts as a structural hint that successfully forces the model's internal attention mechanism to align with the micro-region $R$ (see Section 5.2). Crucially, as shown in our experiments (Section 4), this alignment indeed successfully generalizes to cases where $B$ is removed.

Generally, by distilling the teacher's regional-view expertise into the student's global-view predictions, the student learns to gaze at critical evidence from the full image that would otherwise be overlooked (just like "zoom-in" in mind), thereby internalizing the benefits of zooming while retaining single-pass inference efficiency. More implementation details (box proposal, filtering thresholds, teacher choices, and prompting) are deferred to Appendix B.

### 3.2. ZoomBench

**Construction.** Existing multimodal fine-grained perception benchmarks (Tong et al., 2024; Wang et al., 2025b;f; Wu & Xie, 2024; Zhang et al., 2025c;g) often suffer from limited task coverage, overly templated question styles, low difficulty, or expensive manual construction. To pave the way for building a high-quality (diverse, challenging, and easy-to-scale) benchmark at low cost, we repurpose our *Region to Image Distillation* method. Concretely, we utilize a powerful MLLM to propose questions and candidate answers based on cropped micro regions, then mapped to the full images to form challenging perception tasks *without any spatial grounding (e.g., bounding boxes)*. The cropped regions automatically serves as the evidence regions to answer the questions from the full images. After that, we incorporate a human-in-the-loop verification stage: human annotators are tasked only with checking the validity, difficulty, and correctness of the model-generated question-answer pairs against both the full images and cropped regions. This hybrid approach significantly reduces the labor-intensive burden of from-scratch labeling (creating questions and answers) while maintaining high quality.

*Table 1.* Comparison of *ZoomBench* with other fine-grained multimodal perception benchmarks. "MCQ" denotes multiple-choice-question and "OQ" means open question. Difficulty (higher is better) is measured as "1 − accuracy of Qwen2.5-VL-7B" on each benchmark. Our benchmark emphasizes hybrid high-veracity collection, automatic evidence annotation, diverse challenging questions, along with two unique evaluation protocols.

| Benchmarks | Data Construction | | | | | Evaluation Protocol | |
|---|---|---|---|---|---|---|---|
| | Question Collection | Question Format | Evidence Annotation | Dimension | Difficulty | Dual-View | Interpretability |
| CV-Bench | Manual, Templated | MCQ | ✗ | 4 | 24.7 | ✗ | ✗ |
| VStar | Manual, Templated | MCQ | ✗ | 2 | 19.9 | ✗ | ✗ |
| HR-Bench | Hybrid, Templated | MCQ | ✗ | 6 | 29.6 | ✗ | ✗ |
| MME-RealWorld | Manual | MCQ | ✗ | 10+ | 37.4 | ✗ | ✗ |
| TreeBench | Hybrid | MCQ | Manual | 10 | 63.0 | ✗ | ✗ |
| FINERS-4k | Manual | MCQ, OQ | ✗ | 4 | 33.0 | ✗ | ✗ |
| **ZoomBench (Ours)** | **Hybrid (R2I)** | **MCQ, OQ** | **Auto** | 6 | 57.5 | ✓ | ✓ |

**Statistics.** The resulting *ZoomBench* comprises 845 diverse and challenging QA pairs with high-resolution images sourced from diverse image datasets (detailed in Appendix B.1 and Appendix B.2). We cluster these questions and find that they fall into six popular and major categories in fine-grained perception: (1) Fine-Grained Counting: targeting small and densely packed objects; (2) OCR: focusing on the text and symbol recognition; (3) Color Attributes: discerning subtle color variations in parts of the object; (4) Structural Attributes: examining geometric and shape-related properties such as object structure and part layout; (5) Material Attributes: recognizing material composition and surface properties (e.g., metal, wood, glass, fabric); and (6) Object Identification: distinguishing specific object types and species, such as flags, brands, landmarks, and notable figures.

**Evaluation Protocols.** Our benchmark adopts a hybrid evaluation format that combines multiple-choice questions (for cases where answer spaces are naturally discrete or where strict normalization is hard) and open questions with canonical target answers (please see Appendix C.3 for details of the scoring system). Moreover, *ZoomBench* provides both the full image and the corresponding key-region crop for each instance, which supports deeper analysis in the subsequent section (Section 5). These include the proposed dual-view protocol to measure the "zooming" gap, and attention-map-based (Khayatkhoei et al., 2025) study to interpretably demonstrate the internalized zooming ability. Table 1 further distinguishes the difference between *ZoomBench* and other existing ones. We also provide a data statics demonstration in Figure 8, construction details in Appendix B.1, and several examples of our benchmark in Appendix E.2.

## 4. Experiments

In this section, we describe the experimental settings and present main experimental results. We aim to evaluate the effectiveness of our proposed *Region-to-Image Distillation*

method, focusing on whether our model can internalize zooming expertise to achieve fine-grained perception in a single forward pass.

### 4.1. Experimental Settings

**Benchmarks.** We consider 3 groups of benchmarks. *(i) General perception:* HR-Bench (Wang et al., 2025f) (4K and 8K), VStar (Wu & Xie, 2024), CV-Bench (Tong et al., 2024), MME-RealWorld (Zhang et al., 2025g) (EN and CN), and our *ZoomBench* (full image setting). *(ii) Specific perception:* ColorBench (Liang et al., 2025) for color understanding and robustness, and CountQA (Tamarapalli et al., 2025) for counting in the wild. *(iii) Out-of-distribution (OOD) generalization:* MMStar (Chen et al., 2024) for general multimodal understanding, and BabyVision (Chen et al., 2026a) for visual reasoning.

**Model Training.** We train Qwen3-VL-4B (Bai et al., 2025a), Qwen2.5-VL-7B (Bai et al., 2025b), and Qwen3-VL-8B with reinforcement learning (DAPO (Yu et al., 2025a)) only on our 37k synthetic data (see Appendix C for details) without any SFT. We denote the resulting models as ZwZ (*Zooming without Zooming*), highlighting that they perform fine-grained perception from the full image in a single *glance*.

**Baselines.** We compare against 6 kinds of baselines: **(a)** *closed-source frontier MLLMs*, including GPT-5.1 (OpenAI, 2025) and Gemini-3-Flash (Google, 2025); **(b)** *strong open-source MLLMs*, including Qwen3-VL-235B (Bai et al., 2025a), GLM-4.5V (Hong et al., 2025b), Kimi-K2.5 (Team et al., 2026), MiniCPM-4.5-V (Yu et al., 2025b), and MiMo-VL-7B-RL (Li et al., 2025b); **(c)** *"Thinking-with-Images" agentic models*, including Pixel-Reasoner (Su et al., 2025), DeepEyes (Zheng et al., 2025b), DeepEyesV2 (Hong et al., 2025a), Thyme (Zhang et al., 2025h), Mini-o3 (Lai et al., 2025), SenseNova-MARS (Chng et al., 2025), and Skywork-R1V4 (Zhang et al., 2025e); **(d)** *Qwen3-VL with official tool use*, implemented following the tool-use recipes provided in the official Qwen3-VL Github's cookbook;

*Table 2.* Main results on various benchmarks. We report accuracy (%) for each model. Among open-source models (except GPT-5.1 and Gemini-3-Flash), the best results are highlighted in **bold**, and the second-best are underlined. ZwZ consistently improves over the corresponding Qwen-VL baselines, achieving the best overall average among open-source models.

| Models | General Perception | | | | | | | | Specific Perception | | OOD Generalization | | Avg |
|---|---|---|---|---|---|---|---|---|---|---|---|---|---|
| | ZoomBench | HR-4K | HR-8K | VStar | CV-B. | MME-RW-en | MME-RW-cn | GP-Avg | CountQA | ColorB. | MMStar | BabyVision | |
| *Closed-Source Models* | | | | | | | | | | | | | |
| GPT-5.1 | 47.22 | 67.00 | 65.25 | 70.16 | 84.22 | 64.04 | 55.57 | 64.78 | 31.41 | 83.43 | 71.60 | 13.92 | 59.44 |
| Gemini-3-Flash | 59.29 | 87.88 | 85.00 | 86.39 | 89.57 | 74.86 | 72.62 | 79.37 | 66.88 | 85.47 | 83.60 | 34.51 | 75.10 |
| *Open-Source Models* | | | | | | | | | | | | | |
| Qwen3-VL-4B | 40.24 | 78.25 | 72.88 | 80.10 | 84.95 | 63.47 | 63.63 | 69.07 | 28.14 | 81.63 | 69.73 | 13.66 | 61.52 |
| Qwen2.5-VL-7B | 42.49 | 71.62 | 67.88 | 78.53 | 75.34 | 60.80 | 58.30 | 64.99 | 18.91 | 76.36 | 61.93 | 12.89 | 56.82 |
| Qwen3-VL-8B | 37.87 | 78.88 | 74.63 | 86.39 | 85.44 | 65.96 | 66.67 | 70.83 | 28.99 | 82.77 | 70.93 | 12.89 | 62.86 |
| MiMo-VL-7B-RL | 45.09 | 74.38 | 72.88 | 81.15 | 84.31 | 63.40 | 59.78 | 68.71 | 28.27 | 82.80 | 73.53 | 16.24 | 61.98 |
| MiniCPM-V-4.5 (9B) | 42.60 | 69.88 | 63.62 | 70.16 | 80.25 | 58.16 | 56.23 | 62.99 | 23.43 | 79.75 | 67.87 | 14.95 | 56.99 |
| GLM-4.5V (108B) | 49.23 | 81.63 | 74.88 | 83.25 | 87.59 | 66.04 | 60.71 | 71.90 | 35.93 | 84.59 | 75.87 | 15.72 | 65.04 |
| Qwen3-VL-235B-A22B | 49.11 | **84.50** | 81.62 | 87.96 | 86.72 | 67.07 | 65.29 | 74.61 | 40.58 | 85.62 | 76.33 | 18.30 | 67.55 |
| Kimi-K2.5 (1T) | 56.33 | 81.87 | 75.38 | 85.86 | **89.18** | **71.51** | 68.40 | 75.50 | **52.81** | **86.61** | **81.80** | **33.25** | **71.18** |
| *Our Models* | | | | | | | | | | | | | |
| **ZwZ-4B (Ours)** | 55.74 | 81.75 | 79.50 | **92.67** | 87.90 | 68.52 | 68.09 | 76.31 | 30.82 | 83.08 | 71.13 | 16.24 | 66.86 |
| **ZwZ-7B (Ours)** | 55.62 | 75.38 | 73.25 | 88.48 | 79.83 | 66.21 | 66.96 | 72.25 | 20.72 | 80.82 | 63.40 | 15.98 | 62.42 |
| **ZwZ-8B (Ours)** | **58.11** | 84.38 | **82.00** | 91.10 | 87.40 | 69.87 | **70.59** | **77.64** | 32.40 | 83.59 | 73.13 | 16.75 | 68.12 |

*Table 3.* Comparison of different datasets. The best results are highlighted in **bold**, and the second-best are underlined. Model trained on our synthetic dataset achieves superior performance.

| Training Data | Size | Synthetic? | General Perception | | | | | | | Specific Perception | | OOD Generalization | | Avg |
|---|---|---|---|---|---|---|---|---|---|---|---|---|---|---|
| | | | ZoomBench | HR-4K | HR-8K | VStar | CV-B. | MME-RW-en | MME-RW-cn | CountQA | ColorB. | MMStar | BabyVision | |
| Qwen3-VL-8B | - | - | 37.87 | 78.88 | 74.63 | 86.39 | 85.44 | 65.96 | 66.67 | 28.99 | 82.77 | 70.93 | 12.89 | 62.86 |
| + DeepEyes data | 47K | ✗ | 45.80 | 84.75 | 80.12 | 88.48 | 88.62 | 68.71 | 71.03 | 30.56 | 82.94 | 72.67 | 10.57 | 65.84 |
| + Thyme-RL data | 55K | ✗ | 40.93 | 82.75 | 78.05 | 91.62 | 88.29 | 71.14 | 71.17 | 30.96 | 83.98 | 74.87 | 13.66 | 66.13 |
| + Oasis data | 500K | ✓ | 37.51 | 81.50 | 77.62 | 83.77 | 87.07 | 64.96 | 70.26 | 27.23 | 81.78 | 70.53 | 13.40 | 63.24 |
| + MM-Self-Instruct data | 65K | ✓ | 37.04 | 81.38 | 78.75 | 82.72 | 86.32 | 68.02 | 67.48 | 28.73 | 83.13 | 72.00 | 15.21 | 63.71 |
| **+ our data** | 10K | ✓ | 52.90 | 82.88 | 81.38 | 91.62 | 87.78 | 68.43 | 69.63 | 33.97 | 83.19 | 72.53 | 16.75 | 67.37 |
| **+ our data** | 37k | ✓ | 58.11 | 84.38 | 82.00 | 91.10 | 87.40 | 69.87 | 70.59 | 32.40 | 83.59 | 73.13 | 16.75 | 68.12 |

**(e)** models trained on existing open-sourced multimodal datasets, including synthetic data (Oasis (Zhang et al., 2025b), Multimodal-Self-Instruct (Zhang et al., 2024)) as well as human-curated data (DeepEyes (Zheng et al., 2025b) and Thyme-RL (Zhang et al., 2025h)).

### 4.2. Results

**Leading Performance across Various Benchmarks.** As shown in Table 2, training on our data yields consistent gains on all benchmarks, not limited to perception tasks. Notably, ZwZ-8B improves the Qwen3-VL-8B baseline from 62.86 to 68.12 average score with large gains on *ZoomBench* (37.87 → 58.11), and similar improvements are also observed for ZwZ-4B/7B with different scales. On the average of general **perception benchmarks**, ZwZ-4B and ZwZ-8B also surpass all open-source models including GLM-4.5V, Qwen3-VL-235B, and Kimi-K2.5 with much larger sizes, and remains competitive with the SOTA closed-source model Gemini-3-Flash. This clearly indicates that our data effectively distills the expertise of teacher models (GLM-4.5V and Qwen3-VL-235B) at *cropped region* via zooming to student models (ZwZ), and even enables the

students to outperform the teachers when given the *full image* as global input. Beyond perception-centric benchmarks, ZwZ also reveals strong out-of-distribution performance on MMStar and BabyVision, indicating that learning to reliably retrieve micro evidence from cluttered global inputs benefits broader multimodal understanding and visual reasoning, rather than overfitting to narrow perception patterns.

**Surpassing Existing Open Datasets.** To isolate the contribution of our *Region-to-Image* distilled supervision, we train the same backbone (Qwen3-VL-8B) on several representative public datasets, including human-curated non-synthetic data (DeepEyes, Thyme-RL) and large-scale synthetic data (Oasis, MM-Self-Instruct). Table 3 shows that our distilled samples (10K, 37k) demonstrate strong data efficiency and outperform all alternatives, including datasets that are an order of magnitude larger (e.g., Oasis 500K), suggesting that the key factor is *high quality and fine granularity* rather than sheer data volume.

**Exceeding "Thinking-with-Images" Models.** We further compare ZwZ with representative agentic baselines that explicitly zoom-in during inference. Table 4 shows that our single-pass models are competitive with, and often surpass,

*Table 4.* We compare our models (single forward pass) with agentic models on several perception benchmarks. The best results are highlighted in **bold**, and the second-best are underlined.

| Models | VStar | HR-4K | HR-8K | MME-RW-en | Avg |
|---|---|---|---|---|---|
| *Base Models (Single Forward Pass)* | | | | | |
| Qwen3-VL-4B | 80.1 | 78.3 | 72.9 | 63.5 | 73.7 |
| Qwen2.5-VL-7B | 78.5 | 71.6 | 67.9 | 58.3 | 69.1 |
| Qwen3-VL-8B | 86.4 | 78.9 | 74.6 | 66.0 | 76.5 |
| *"Thinking-with-Images" Agentic Models* | | | | | |
| Pixel-Reasoner (7B) | 84.3 | 72.6 | 66.1 | 64.4 | 71.9 |
| DeepEyes (7B) | 85.6 | 75.1 | 72.6 | 64.1 | 74.4 |
| Thyme (7B) | 82.2 | 77.0 | 72.0 | 64.8 | 74.0 |
| DeepEyesV2 (7B) | 81.8 | 77.9 | 73.8 | 64.9 | 74.6 |
| Mini o3 (7B) | 88.2 | 77.5 | 73.3 | 65.5 | 76.1 |
| SenseNova-MARS-8B | 92.2 | 83.1 | 78.4 | 67.9 | 80.4 |
| Skywork-R1V4 (30B) | 88.0 | 82.8 | 79.8 | **71.4** | 80.5 |
| *Qwen3-VL with Official Tool Use* | | | | | |
| Qwen3-VL-4B + tool | 88.0 | 81.3 | 74.4 | 65.4 | 77.3 |
| Qwen3-VL-8B + tool | 90.1 | 82.3 | 78.0 | 66.0 | 79.1 |
| *Our Models (Single Forward Pass)* | | | | | |
| **ZwZ-4B** | **92.7** | 81.8 | 79.5 | 68.5 | 80.6 |
| **ZwZ-7B** | 88.5 | 75.4 | 73.3 | 66.2 | 75.9 |
| **ZwZ-8B** | 91.1 | **84.4** | **82.0** | 69.9 | **81.9** |

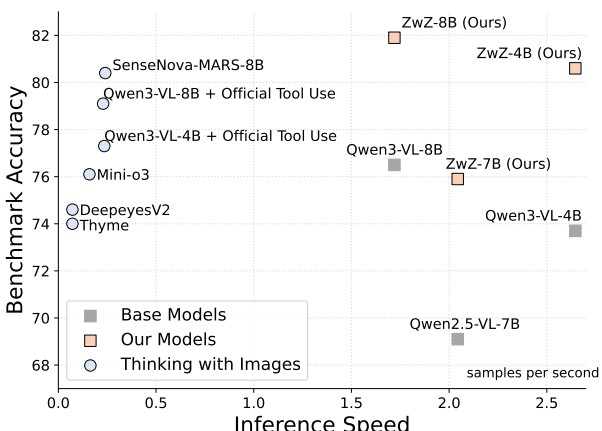

*Figure 4.* Benchmark accuracy versus inference speed. Our models achieve higher accuracy than base models and agentic baselines while retaining single-pass efficiency.

strong agentic systems. In particular, ZwZ-8B achieves the best overall average among the listed methods, demonstrating that our approach can exceed the accuracy benefits of inference-time zooming while avoiding iterative tool calls.

**Internalizing Tool-Use Benefits with Lower Latency.** We also implement the official Qwen3-VL tool-use pipeline and compare against it (Table 4). ZwZ surpasses the tool-use counterpart while using only a single forward pass, directly suggesting that *Region-to-Image Distillation* effectively *internalizes* the gains of tool-based zooming into model weights. Moreover, Figure 4 reports the accuracy–speed trade-off, where inference speed is defined as the inverse of the average per-sample inference time on *ZoomBench*, and benchmark accuracy is the average score in Table 4. ZwZ attains higher accuracy with substantially lower inference cost (around 10× faster inference speed) than agentic and tool-use baselines.

### 4.3. Ablation Study

In this section, we conduct a series of ablation experiments on Qwen3-VL-8B to investigate the impact of key components in our Region-to-Image (R2I) distillation framework. Note that we use 10K data for efficient comparison in our ablation study.

**Effectiveness of Region-to-Image Distillation.** We first compare our *Region-to-Image Distillation* method with the "Direct Synthesis" method, which nearly represents the traditional "global-to-global" synthesis and "teacher model to student model" distillation approaches (e.g., Genixer (Zhao

et al., 2024), MM-Evol (Luo et al., 2025), Oasis (Zhang et al., 2025b)). In this setting, the teacher model is prompted with the full image to generate VQA pairs to train the student model. As shown in Figure 5, our method (R2I + bbox-in-image) significantly outperforms Direct Synthesis across all benchmarks, which confirms that generating supervision on micro-crops effectively bypasses the hallucinatory tendencies of teacher models when facing global views, thereby providing higher-quality training signals.

**Importance of Visual Grounding.** A critical challenge in R2I is the referential ambiguity when distilling crop-derived answers back to the full image. One straightforward way is designing an agentic workflow to verify whether each synthesized QA pair is strictly unambiguous at the full-image level. However, this per-sample validity checking is costly and would also filter out a substantial portion of the data. An alternative is to query the MLLM to rewrite each question conditioned on the full image to remove ambiguity. However, in our manual inspection, many rewritten questions introduce severe hallucinations. Thus, we ablate three grounding strategies (including our final chosen strategy): (a) R2I + no-bbox: Directly distilling the crop-based VQA pairs to the full image without any spatial guidance. (b) R2I + bbox-in-question: Providing the target bounding box coordinates as text (e.g., $[x_1, y_2, x_2, y_2]$) within the prompt. (c) R2I + bbox-in-image (ours): Overlaying the bounding box directly onto the image. As illustrated in Figure 5, both the no-bbox and bbox-in-question setting performs worse than the bbox-in-image format. We posit that overlaying boxes on images acts as a strong *privileged information*. Besides its original purpose to resolve referential uncertainty, it also effectively forces the model's visual encoder and cross-modal projector to learn a direct mapping between the global context and the specific micro-region. By contrast,

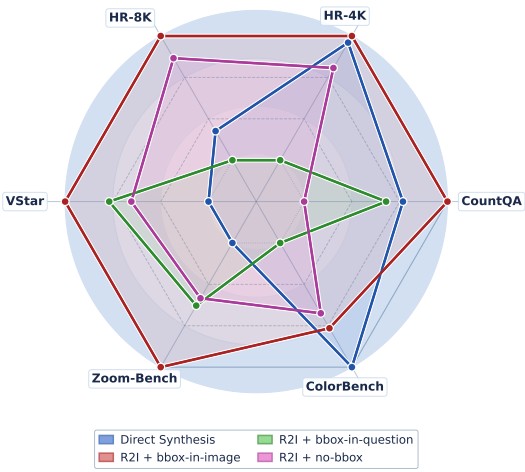

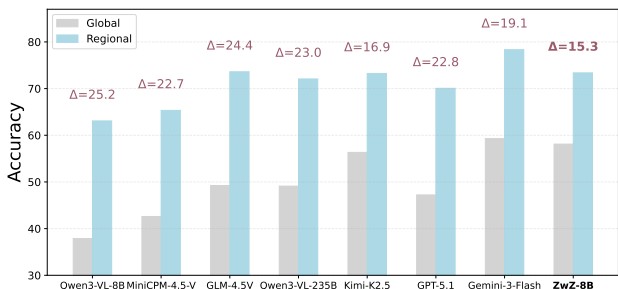

*Figure 7.* Dual-view evaluation. ZwZ-8B exhibits strong performance on both "Global-View" and "Regional-View" with the smallest zooming gap.

*Figure 5.* Ablation study using different data synthesis strategies.

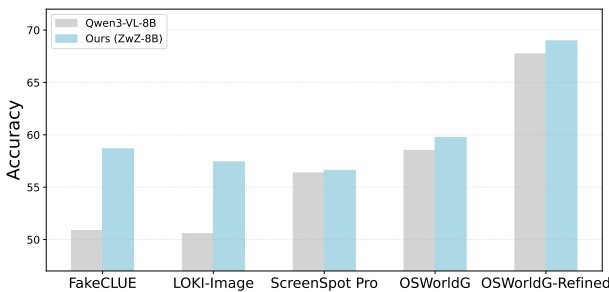

*Figure 6.* Generalization to real-world tasks. ZwZ-8B consistently improves over Qwen3-VL-8B on AIGC detection and GUI agent benchmarks.

textual coordinates must first be parsed and then aligned back to visual tokens, requiring the LLM to 'reason' about which visual tokens correspond to which numbers, which introduces an additional burden. In this sense, the bbox overlay on the image serves as a stronger signal during training, making it easier for the model to learn the correspondence between the global image and the relevant micro region. Crucially, although these boxes are only present during training, our results (where no boxes are provided at test time) demonstrate that the model effectively internalizes this "zoom-in" capability, achieving superior fine-grained perception in a single "*glance*".

### 4.4. Real World Tasks Generalization.

To evaluate broader applicability, we further test on real-world tasks that require strong multimodal grounding, including AIGC detection (LOKI (Ye et al., 2025), Fake-CLUE (Wen et al., 2025)) and GUI agents (ScreenSpot Pro (Li et al., 2025c) and OSWorldG (Xie et al., 2025)). As shown in Figure 6, ZwZ-8B consistently outperforms Qwen3-VL-8B across these benchmarks. These results

demonstrate that our approach enhances not only fine-grained perception but also foundational multimodal capabilities across diverse real-world applications.

## 5. Deeper Analysis on ZoomBench

Going beyond standard benchmarking, we leverage *ZoomBench* to more carefully diagnose fine-grained perception under two additional evaluation protocols.

### 5.1. Dual-View Evaluation

Motivated by Khayatkhoei et al. (2025), we introduce a *dual-view* evaluation protocol to explicitly measure the "zooming" gap in *ZoomBench*. Each instance is evaluated in two conditions: (1) *Global-View*, where the model answers from the full image; and (2) *Regional-View*, where the model is given the corresponding micro-cropped region. The Regional-View accuracy serves as an empirical upper bound (whether the model can answer when the evidence is explicitly visible), while the Global-View accuracy measures whether the model can *find and use* the same evidence from the full image. We define the *zooming gap* as the Global–Regional performance difference; a larger gap indicates that the sample is solvable given the evidence, yet the model fails to reliably pay attention to it under realistic global inputs, which is precisely the fine-grained perception (not recognition) bottleneck targeted by *ZoomBench*.

From Figure 7 and Table 5, we observe a consistent and substantial zooming gap for vanilla MLLMs across scales. For example, Qwen3-VL-8B drops from 63.08% (Regional) to 37.87% (Global), yielding a 25.21% gap, which suggests that many failures stem from perceptual oversight rather than insufficient recognition or reasoning. Moreover, even much larger models (e.g., GLM-4.5V, Qwen3-VL-235B, Kimi-K2.5, and GPT-5.1) still exhibit notable gaps, indicating that scaling parameters alone does not eliminate the fine-grained retrieval bottleneck. In contrast, ZwZ-8B largely narrows this gap to only 15.26%, smallest among the tested

*Table 5.* Detailed dual-view results on *ZoomBench*. We report Global-View, Regional-View, and their difference (*Zooming Gap*) across six perceptual dimensions. Note that row-wise averages are weighted by the number of samples in each dimension, since different dimensions contain different amounts of data. For the averaged scores, the best are highlighted in **bold**, and the second-best are underlined.

| Models | Dual-View | ZoomBench | | | | | | Avg |
|---|---|---|---|---|---|---|---|---|
| | | Counting | OCR | Color | Structure | Material | Identification | |
| Qwen3-VL-8B | Global | 29.90 | 47.97 | 38.02 | 37.42 | 39.34 | 43.33 | 37.87 |
| | Regional | 48.04 | 69.11 | 64.05 | 74.19 | 63.93 | 68.33 | 63.08 |
| | *Zooming Gap* | 18.14 | 21.14 | 26.03 | 36.77 | 24.59 | 25.00 | 25.21 |
| MiniCPM-4.5-V | Global | 31.86 | 47.97 | 44.63 | 46.45 | 49.18 | 43.33 | 42.60 |
| | Regional | 45.10 | 78.05 | 67.77 | 71.61 | 75.41 | 71.67 | 65.33 |
| | *Zooming Gap* | 13.24 | 30.08 | 23.14 | 25.16 | 26.23 | 28.34 | 22.73 |
| GLM-4.5V | Global | 32.84 | 63.41 | 50.41 | 52.90 | 47.54 | 63.33 | 49.23 |
| | Regional | 57.84 | 78.05 | 74.79 | 81.29 | 80.33 | 86.67 | 73.61 |
| | *Zooming Gap* | 25.00 | 14.64 | 24.38 | 28.39 | 32.79 | 23.34 | 24.38 |
| Qwen3-VL-235B | Global | 37.75 | 61.79 | 46.69 | 58.06 | 42.62 | 55.00 | 49.11 |
| | Regional | 58.33 | 82.11 | 71.49 | 78.71 | 77.05 | 78.33 | 72.07 |
| | *Zooming Gap* | 20.58 | 20.32 | 24.80 | 20.65 | 34.43 | 23.33 | 22.96 |
| Kimi-K2.5 | Global | 44.61 | 67.48 | 53.31 | 62.58 | 65.57 | 60.00 | 56.33 |
| | Regional | 62.25 | 80.49 | 71.90 | 79.35 | 83.61 | 75.00 | 73.25 |
| | *Zooming Gap* | 17.64 | 13.01 | 18.59 | 16.77 | 18.04 | 15.00 | 16.92 |
| GPT-5.1 | Global | 34.80 | 54.47 | 48.76 | 49.03 | 57.38 | 53.33 | 47.22 |
| | Regional | 52.45 | 78.86 | 71.49 | 78.71 | 77.05 | 76.67 | 70.06 |
| | *Zooming Gap* | 17.65 | 24.39 | 22.73 | 29.68 | 19.67 | 23.34 | 22.84 |
| Gemini-3-Flash | Global | 42.16 | 80.49 | 54.55 | 61.94 | 72.13 | 73.33 | **59.29** |
| | Regional | 68.14 | 89.43 | 74.38 | 87.10 | 81.97 | 80.00 | **78.34** |
| | *Zooming Gap* | 25.98 | 8.94 | 19.83 | 25.16 | 9.84 | 6.67 | 19.05 |
| **ZwZ-8B** | Global | 40.69 | 67.48 | 68.60 | 58.71 | 57.38 | 55.00 | 58.11 |
| | Regional | 54.90 | 81.30 | 81.40 | 76.13 | 78.69 | 75.00 | 73.37 |
| | *Zooming Gap* | 14.21 | 13.82 | 12.80 | 17.42 | 21.31 | 20.00 | 15.26 |
| **Average** | Global | 36.83 | 61.38 | 50.62 | 53.39 | 53.89 | 55.83 | 49.97 |
| | Regional | 55.88 | 79.67 | 72.16 | 78.39 | 77.25 | 76.46 | 71.14 |
| | *Zooming Gap* | 19.05 | 18.29 | 21.54 | 25.00 | 23.36 | 20.63 | 21.17 |

*Table 6.* Attention map bounding-box coverage on *ZoomBench*. Higher is better.

| Models | QwenVL-4B | QwenVL-7B | QwenVL-8B | ZwZ-4B | ZwZ-7B | ZwZ-8B |
|---|---|---|---|---|---|---|
| **Coverage (%)** | 17.34 | 12.44 | 17.39 | **21.45 (4.11↑)** | 13.39 (0.95↑) | 21.64 (4.25↑) |

*tion* (Khayatkhoei et al., 2025) falls inside the annotated key-region bounding box for each VQA sample in *ZoomBench*. For each image-question pair, we compute a relative attention map $A_{\mathrm{rel}}$ over the visual token grid (see Appendix F for details). Note that in *ZoomBench*, each sample is annotated with a ground-truth bounding box $B = (x_1, y_1, x_2, y_2)$ on the original image, indicating the minimal key region required to answer the question. Thus, we map the box to the visual token grid to obtain $\tilde{B}$, and then define the *Relative-Attention Coverage Ratio* as the fraction of total relative-attention mass inside the key region:

$$\mathrm{Coverage}(B) = \frac{\sum_{(i,j) \in \tilde{B}} A_{\mathrm{rel}}(i,j)}{\sum_{(i,j)} A_{\mathrm{rel}}(i,j)}. \qquad (2)$$

A higher coverage value indicates that the model's question-relevant visual evidence is more concentrated within the annotated key region.

**Results.** Table 6 reports the average bounding-box coverage of relative-attention mass on *ZoomBench*. Across all model scales, our ZwZ variants consistently achieve higher coverage than their corresponding Qwen3-VL baselines, indicating that the model's question-relevant attention is more concentrated within the annotated key region. This also suggests that our method enhances the model's ability to *find (pay "attention" to) and utilize* micro-level evidence during inference, and thereby narrowing the Global–Regional "zooming" gap discussed in Section 5.1. We also present some demonstrations of attention map comparison in Appendix F.

## 6. Conclusion

We propose Zooming without Zooming, a *Region-to-Image Distillation* framework that converts inference-time zooming into a training-time primitive. By synthesizing high-veracity VQA supervision on micro-crops and distilling it back to full images with explicit grounding, our approach teaches MLLMs to recover fine-grained evidence from global inputs in a single forward pass, avoiding the latency of tool-based "Thinking-with-Images" inference. We also introduce *ZoomBench* with a dual-view protocol that quantifies the global–regional zooming gap. Experiments show consistent gains in fine-grained perception, outperforming agentic baselines at much lower cost, while also improving general multimodal cognition, yielding notable gains on visual reasoning and real-world tasks. Analyses further confirm that our models attend better to task-relevant micro regions and substantially narrow the zooming gap.

models. It also achieves the second highest Global-View accuracy (58.11%) that nearly rivals the Regional-View performance of the vanilla Qwen3-VL-8B baseline.

A dimension-wise analysis of Table 5 further reveals several key insights: (1) Counting emerges as the most challenging dimension. Unlike other dimensions where Regional-View substantially improves performance, Counting remains difficult even after zooming in, suggesting that the task requires not only precise localization of densely packed objects but also sophisticated reasoning to avoid over- or under-counting. (2) The largest average zooming gaps appear in Structure and Material, suggesting that subtle surface/texture attributes are especially prone to attention dilution under global inputs. (3) In contrast, OCR and Identification show smaller gaps, implying these more salient, discrete cues are comparatively easier to exploit.

Generally, these results demonstrate that our *Region-to-Image Distillation* effectively teaches the model to focus on the "needle in the haystack" directly from the global view and effectively bridges the gap between global and regional perception without requiring test-time tool calling.

### 5.2. Attention Map Coverage Analysis

To further evaluate whether the model grounds its predictions on task-relevant image regions from a view of interpretability, we measure how much of the *Relative Atten-*

## Acknowledgements

Weiran Huang is supported by National Natural Science Foundation of China (No. 62406192), Shanghai Municipal Special Program for Basic Research on General AI Foundation Models (Grant No. 2025SHZDZX025G03), Tencent WeChat Rhino-Bird Focused Research Program, and Kuaishou Technology. This work is supported by Ant Group Research Intern Program. This work is also supported by the Zhongguancun Academy (Grant No.s C20250203).

## Impact Statement

This paper presents work whose goal is to advance the field of Machine Learning. There are many potential societal consequences of our work, none which we feel must be specifically highlighted here.

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

# Appendix

## A. Discussion and Future Direction

---

**Algorithm 1** *Region-to-Image Distillation* (R2I)

---

**Require:** Raw image pool $\mathcal{D}_{\text{raw}}$; proposal function $\mathcal{P}(\cdot)$; question generator $\mathcal{T}_{\text{gen}}$; teacher ensemble $\{\mathcal{T}_1, \ldots, \mathcal{T}_m\}$; sparsity threshold $\tau$; max questions per crop $K$
**Ensure:** Distilled dataset $\mathcal{D}_{\text{syn}}$
1: $\mathcal{D}_{\text{syn}} \leftarrow \emptyset$        // initialize
2: **for each** image $I \in \mathcal{D}_{\text{raw}}$ **do**
3:    $\mathcal{B} \leftarrow \mathcal{P}(I)$        // propose candidate boxes via detection/segmentation
4:    **for each** box $B \in \mathcal{B}$ and $\text{AREA}(B)/\text{AREA}(I) \leq \tau$ **do**
5:      $R \leftarrow \text{CROP}(I, B)$        // zoom in: crop region $R$ from $I$
6:      $\mathcal{Q}_R \leftarrow \mathcal{T}_{\text{gen}}(R; K)$        // generate $K$ region-answerable questions
7:      **for each** question $Q \in \mathcal{Q}_R$ **do**
8:        $\mathbf{a} \leftarrow [\,]$        // collect teacher answers
9:        **for each** teacher $\mathcal{T}_j \in \{\mathcal{T}_1, \ldots, \mathcal{T}_m\}$ **do**
10:         $a_j \leftarrow \mathcal{T}_j(R, Q); \quad \mathbf{a} \leftarrow \mathbf{a} \cup \{a_j\}$        // answer on the crop to reduce hallucination
11:        **end for**
12:        **if** $\text{HIGHCONSENSUS}(\mathbf{a})$ **then**
13:         $A \leftarrow \text{CONSENSUSMAP}(\mathbf{a})$        // e.g., majority vote
14:         $(I', Q') \leftarrow \text{GROUNDTOFULL}(I, B, Q)$    // zoom out: overlay $B$ on $I$ to form $I'$ and add a spatial constraint to form $Q'$
15:         $\mathcal{D}_{\text{syn}} \leftarrow \mathcal{D}_{\text{syn}} \cup \{(I', Q', A)\}$        // store full-image training triplet
16:        **end if**
17:      **end for**
18:    **end for**
19: **end for**
20: $\mathcal{D}_{\text{syn}} \leftarrow \text{REJECTIONSAMPLING}(\mathcal{D}_{\text{syn}})$
21: **return** $\mathcal{D}_{\text{syn}}$

---

**Algorithm 2** *A general view of our method.*

---

**Require:** Raw image pool $\mathcal{D}_{\text{raw}}$; a tool-call action $f(\cdot)$; question-answer generator $\mathcal{T}$
**Ensure:** Distilled dataset $\mathcal{D}_{\text{syn}} = \{(I, \widehat{Q}, A)\}$
1: $\mathcal{D}_{\text{syn}} \leftarrow \emptyset$
2: **for each** image $I \in \mathcal{D}_{\text{raw}}$ **do**
3:    $\widehat{I} = f(I)$        // conduct a tool-call action (e.g., "zoom-in")
4:    $(Q, A) \sim \mathcal{T}(\widehat{I})$        // generate question-answer pairs based on $\widehat{I}$
5:    $I, \widehat{Q} = f^{-1}(\widehat{I}, Q)$        // inverse transformation
6:    $\mathcal{D}_{\text{syn}} \leftarrow \mathcal{D}_{\text{syn}} \cup \{(I, \widehat{Q}, A)\}$
7: **end for**
8: **return** $\mathcal{D}_{\text{syn}}$

---

In this section, we first present a general abstraction of our method as a unified framework for tool-action distillation Next, we compare with different recently popular multimodal "thinking" paradigms. Then, we further talk about when to use agentic workflows like "Thinking with Images" and when not to.

### A.1. A General View of Our Method

The overall pipeline of our method is presented in Algorithm 1. From a broader perspective, our method can be further abstracted as *tool-action distillation*, which is illustrated in Algorithm 2. In particular, given a raw image $I$, we first apply a tool-call action $f(\cdot)$ to obtain an altered observation $\widehat{I}$ (e.g., a zoomed-in crop). We then use a teacher generator $\mathcal{T}$ to synthesize question–answer pairs $(Q, A)$ conditioned on $\widehat{I}$, so that the questions explicitly require the visual evidence revealed or emphasized by $f$. Next, we map each question back to the original image domain via an inverse transformation $f^{-1}$, producing a re-anchored question $\widehat{Q}$ such that $(I, \widehat{Q})$ is unambiguous and still refers to the same evidence used to answer $(\widehat{I}, Q)$. This yields a distilled dataset $\mathcal{D}_{\text{syn}} = \{(I, \widehat{Q}, A)\}$ that can train a student to solve the task directly from the full image, thereby internalizing the benefits of the tool action into a single forward pass of the student. Here, $\widehat{Q}$ denotes the original question $Q$ augmented with the tool/action description when necessary (e.g., explicitly stating how we perform the

tool-call action on the image), while in cases without ambiguity we simply have $\widehat{Q} = Q$. Note that in our implementation, for stronger spatial grounding, we further convert $(I, \widehat{Q})$ into $(I', Q')$ by (i) overlaying the bounding box on $I$ to form $I'$, and (ii) appending an explicit constraint prompt to form $Q'$. Although we instantiate $f$ as "zooming in" as it is the most widely used tool among "Thinking with Images" actions, the same framework can distill many other tool actions, such as flipping (Xu et al., 2025), 3D grounding (Chen et al., 2025; Han et al., 2025), and calling expert models (Lu et al., 2025; Zhou et al., 2025). We further discuss this in Section A.3.

### A.2. Comparison of Different Multimodal "Thinking" Paradigms

Currently, there are many works that focus on developing new multimodal "thinking" paradigms beyond standard textual reasoning. We summarize several representative ones as follows.

**(1) Textual CoT with Additional Tags or Structured Traces.** A line of work injects explicit intermediate structures into the reasoning trace, such as special tokens and bounding-box, line coordinates for visual grounding (Chen et al., 2026b; Li et al., 2025e; Yang et al., 2025b; 2026; Yuan et al., 2025; Zhang et al., 2025a; Zhu et al., 2026). While such formats can improve faithfulness, controllability, and clear interpretability on targeted tasks, these training data are often expensive to curate. They also risk *format overfitting*: the model may over-rely on a particular reasoning template that is not universally helpful, raising an adaptivity issue, i.e., the model must learn when to follow the template versus answer directly under out-of-distribution (OOD) inputs.

**(2) "Thinking with Images": Tool-Based Agentic Reasoning.** Agentic TwI methods (Wang et al., 2025d; Zhang et al., 2025h; Zheng et al., 2025b) (we mainly compare with) call external tools (zoom, rotate, detectors, retrievers, etc.) during inference to obtain improved observations or auxiliary signals. They are flexible and can handle open-world settings with clear interpretability, but are typically slower due to multi-step interaction and require solving an adaptive tool-use problem: deciding *whether*, *when*, and *how* to invoke tools. A mismatched or overly-used tool policy can degrade performance in many scenarios.

**(3) "Thinking with Images": Native Image Generation.** Another direction focuses on developing unified multimodal models (Deng et al., 2025; Li et al., 2025a; Zhang et al., 2025i) or world models (Wiedemer et al., 2025; Zhao et al., 2025) that use native image generation as a scratchpad (e.g., drawing or editing images) during reasoning. This mutlimodal reasoning paradigm is also called "thinking with generating images". This can provide additional visual cues, but inference is very slow (image generation is expensive) and training typically needs paired interleaved image-text interaction data, making large-scale data construction challenging. Moreover, current unified models (e.g., Bagel (Deng et al., 2025)) still lag behind state-of-the-art textual-CoT-based multimodal LLMs (e.g., Qwen3-VL (Bai et al., 2025a)) in multimodal understanding tasks.

**(4) "Thinking with Images": Implicit Latent Reasoning.** To solve the slow inference problem in "thinking with generating images", many implicit approaches replace explicit image-generation steps with latent-space multimodal reasoning (Dong et al., 2025; He et al., 2025; Wang et al., 2025e; 2026; Wu et al., 2026; Yang et al., 2025c). Though promising, currently shifting a pretrained model from default textual reasoning to latent reasoning (generating images in the latent space) typically requires large-scale and diverse interleaved image-text post-training data, which can be costly. This is similar to what is needed to train unified models, and also faces an adaptivity issue to avoid harming OOD performance. In addition, this reasoning paradigm also lacks clear visual interpretability, and current models trained in the paradigm remain behind strong multmodal LLM baselines. For instance, Monet-7B (using latent CoT) reports 71.0 on HR-Bench-4K and 83.3 on VStar, compared with DeepEyes (tool-base agenetic model; 75.1 / 90.1) and ZwZ-7B (text-only CoT model; 75.4 / 88.5) under the same benchmarks.

**(5) Textual CoT with Stronger Perception (Ours).** We summarize the limitations of prior methods along three practical axes: (i) *test-time cost*, (ii) *data/engineering overhead*, and (iii) *adaptivity*. We find that many image-space operations can be *described* and *constrained* in certain language patterns, allowing pure-text reasoning to cover a broad set of perceptual image operations without the need of additional "Thinking with Images" during inference. Therefore, in our implementation, we choose to maintain the simplest and the most widely used paradigm, i.e., textual CoT, and focus on improving perception under single-pass inference via reinforcement learning. This post-training recipe is simple, efficient and scalable: (1) synthetic VQA pairs are generated from images alone, making data construction largely automatic and easy to scale; (2) the model does not rely on SFT for external guidance, thereby maintaining OOD performance; (3) it also guarantees advanced performance after post-training once we get a strong pretrained multimodal model (current SOTA MLLMs are usually based on textual CoT for reasoning, e.g., Qwen3-VL). It achieves a good tradeoff among inference speed, clear interpretability,

*Table 7.* Distillability of common TwI tool/actions under the information-gain criterion. Information-neutral actions are usually predictable based on current image view and the MLLM's strong cognition. Therefore, we argue that these benefits can be *internalized* into the MLLM by training solely on textual CoT via our method (Algorithm 2).

| Category | Tool/Action for Images (examples) | Predictable? |
|---|---|---|
| **Information-gain actions** *(unpredictable external information)* | | |
| Web search / retrieval | Bring in new, unseen images from external sources (Wu et al., 2025a) | ✗ |
| **Information-neutral actions** *(reformat/reveal existing evidence)* | | |
| Zoom in (crop)[1] | Zooming in key regions for easier perception (Zheng et al., 2025b) | ✓ |
| Zoom out | Zooming out for detecting adversarial AIGC (Xu et al., 2025) | ✓ |
| Flip | Horizontal/vertical flipping (Xu et al., 2025) | ✓ |
| Rotate | Rotation for viewpoint normalization (Zhang et al., 2025h) | ✓ |
| Denoise | Remove Gaussian noise (Xu et al., 2025) | ✓ |
| 2D grounding | 2D object detection and segmentation (Lu et al., 2025; Zhou et al., 2025) | ✓ |
| 3D grounding | Depth estimator; 3D object detection and segmentation (Chen et al., 2025) | ✓ |
| Draw images | Draw sketches or auxiliary lines (Wei et al., 2025a; Zhao et al., 2025) | ✓ |

and strong performance.

### A.3. Rethinking "Thinking with Images"

Note that our superior results of *Region-to-Image Distillation* (R2I) on various benchmarks actually do not argue against "Thinking-with-Images" (denoted as TwI). Rather, they suggest a practical criterion (shown in Table 7): TwI is most valuable when the action produces unpredictable *information gain*.

**Information-Gain Actions: TwI Remains Essential.** TwI is well motivated when image actions result to new information. For example, web search or retrieval (Wu et al., 2025a) is essential as it brings in new images or documents. In these cases, the tool's outputs are totally unpredictable from the current view of image and cannot be replaced by purely internalized weights, because the agent *must* interact with an external source to gain more information.

**Information-Neutral Actions: Distill instead of Acting.** In contrast, many common TwI actions concentrate on image editing (e.g., zooming (Zheng et al., 2025b), flipping (Xu et al., 2025), rotating (Zhang et al., 2025h), objection detection and segmentation (Lu et al., 2025; Zhou et al., 2025), drawing auxiliary constructs (Wei et al., 2025a; Wiedemer et al., 2025; Wu et al., 2025b; Zhao et al., 2025)) that are often predictable based on current image view and the MLLM's strong cognition. They typically do not add new information; they mainly reduce perceptual difficulty by making existing evidence easier to access. For such cases, thinking with images at inference time can be inefficient and unnecessary, and the benefits can often be *internalized* into the MLLM (we assume that MLLM has enough knowledge capacity) through training with textual CoT alone, which is exactly what our method (R2I) does (Algorithm 2). Besides, several prior studies (Du et al., 2025; Liu et al., 2025b) have observed that training with pure textual CoT can achieve performance comparable to agentic training when using the same training dataset (Zhang et al., 2025h; Zheng et al., 2025b). We further find that **many agentic TwI models (Zhang et al., 2025h; Zheng et al., 2025b) already achieve strong performance without using any tools, which is comparable to, or even better than, using tools (Table 8)**. These findings also support our argument that such tool use is usually unnecessary.

### A.4. Rethinking Agentic Workflows (Test-Time Scaling)

Diving deeper, we further analyze whether the popular and classical *parallel divide-and-conquer* agentic workflows (e.g., ReAct (Yao et al., 2022)), which is also similar to test-time scaling techniques (Yuksekgonul et al., 2026; Zheng et al., 2026), can also be internalized into a LLM's single forward pass. In our view, if the result of each process in the workflow is exactly a part of the final answer, parallelization primarily improves wall-clock speed and can sometimes reduce cost by delegating simpler processes to cheaper LLMs. It therefore remains practically useful and essential. For example, in

---

[1] Note that zooming in can sometimes yield information gain. When the original image is too high-resolution to be fed into the MLLM directly, it is typically downsampled, which can erase fine details in small regions. Zooming in such regions preserves their effective resolution, recovering visual evidence that is missing in the downsampled global view and thus providing additional information compared to not zooming in. In our data synthesis and training, we encode images at full resolution without downsampling.

*Table 8.* Comparison of DeepEyes and Thyme (both trained with agentic mode) under two inference modes: *agentic mode* (with iterative tool calls) and *direct answering mode* (single forward pass without any tool use). The performance gap between the two modes is surprisingly small, suggesting that the agentic overhead contributes marginally to the final accuracy on these perception benchmarks.

| Models | Inference Mode | VStar | HR-4K | HR-8K | MME-RW-en | Avg |
|---|---|---|---|---|---|---|
| DeepEyes | Agentic | 85.6 | 75.1 | 72.6 | 64.1 | 74.4 |
|  | Direct Answering | 81.2 | 74.6 | 69.9 | 63.9 | 72.4 |
| Thyme | Agentic | 82.2 | 77.0 | 72.0 | 64.8 | 74.0 |
|  | Direct Answering | 82.7 | 77.0 | 71.6 | 62.1 | 73.4 |

agentic systems such as Cursor (Cursor, 2024), Claude Code (Anthropic, 2025), and OpenClaw (OpenClaw, 2026), two agents can work concurrently on frontend (using a cheaper LLM) and backend code (using a more expensive but much stronger LLM), and each process produces an artifact that directly constitutes part of the final deliverable. In contrast, if intermediate outcome of a parallel process does not appear in the final output but merely serves as auxiliary reference (e.g., using agent debate in MoltBook (MoltBook, 2026) to solve problems), then the benefits of such a parallelization workflow can be internalized through training (e.g., DeepSeek-R1 (Guo et al., 2025) internalizes the reflection-based workflow into its own thinking pattern, and MM-UPT (Wei et al., 2025b) internalizes the benefits of majority voting into the model's single forward pass). This also resembles the "System 2 to System 1 Distillation".

### A.5. Failure Case Analysis

We identify three representative failure modes of our method:

- **Global Relational Reasoning.** While R2I excels at localizing and identifying micro-evidence, it can sometimes struggle with tasks requiring long-range relational reasoning between distant objects. The model's enhanced focus on specific regions may not be sufficient to solve cases that demand holistic spatial comparisons across the full image. However, our methodology can be readily extended to this domain by synthesizing training data through spatial reasoning or multi-object searching tools (Chen et al., 2025; Wang et al., 2025f; Wu et al., 2025b) based on Algorithm 2.

- **Teacher-Inherited Hallucinations.** Since our student model learns from multi-teacher consensus, in rare cases where multiple teachers converge on the same incorrect micro-detail, the student may inherit this hallucination. Although our strict consensus filtering ($>6/8$ agreement) substantially mitigates this, it cannot eliminate it entirely.

- **Fundamental Resolution Limits.** When the original image is too high-resolution to be fed into the MLLM directly, it is typically downsampled, which can erase fine details in small regions. In such cases, the decisive evidence may be lost before the model even processes the image, and no amount of training-time zooming supervision can recover it.

### A.6. Future Direction

A promising direction is to develop a unified and dynamic agent policy (Bai et al., 2026; Wang et al., 2025a) for token effiency. Based on our analysis, this agent can (i) default to single-pass inference with enhanced perceptual skills, (ii) decide when and how to invoke tools, and (iii) prioritize information-gain TwI actions, while using information-neutral operations only sparingly. This would combine the efficiency of our method with the open-world capability of agentic TwI.

## B. Implementation Details of Our Method

We present the implementation details of our data synthesis method to construct the high=quality training dataset and *ZoomBench*.

### B.1. Region-to-Image Distillation

To ensure broad visual diversity and high information density, we curate a pool of high-resolution images (mostly over $800 \times 800$ pixels) from several large-scale datasets, including SA-1B (Kirillov et al., 2023), LAION (Schuhmann et al., 2022), MetaCLIP (Chuang et al., 2025), Visual Genome (Krishna et al., 2017), CC12M (Changpinyo et al., 2021), and STPLS3D (Chen et al., 2022). For datasets lacking native object-level annotations, we implement an object recognition

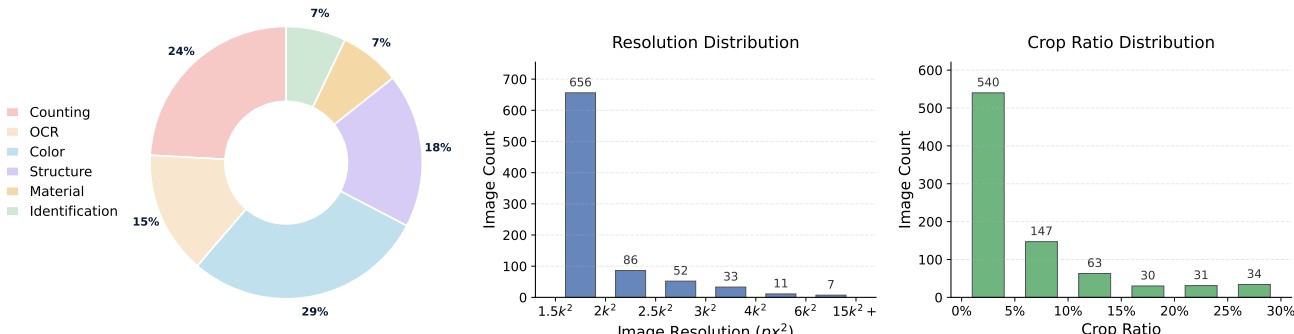

*Figure 8.* Category distribution across six fine-grained dimensions of our benchmark (left) and *ZoomBench* data statistics: distribution of image resolutions (middle) and crop-to-image area ratios (right).

and segmentation system. In particular, we employ Qwen3-VL-235B (Bai et al., 2025a) to generate comprehensive object inventories based on the images, which are subsequently processed by SAM3 (Carion et al., 2025) to produce precise bounding boxes $B$ and cropped regions $R$. To specifically target fine-grained challenges, we filter these candidates to retain only regions $R$ much smaller than the full images $I$, satisfying the area ratio $Area(R)/Area(I) < 0.1$. For data synthesis, we firstly *zoom in* (crop and then resize by $2\times$) to synthesize VQA pairs directly for objects in each cropped region $R$. In particular, we use Qwen3-VL-235B (Bai et al., 2025a) as the question generator, and Qwen3-VL-235B (Bai et al., 2025a) together with GLM-4.5V (Hong et al., 2025b) as two independent answer generators. These two models are strong yet cost-effective for data synthesis. To obtain reliable pseudo-labels, we sample four responses per answer generator (8 in total) and keep a QA pair only when the majority answer reaches a strict consensus ($> 6/8$); otherwise, we discard it. After distilling back to get the augmented vqa $I', Q', A$, we also filter out data that Qwen3-VL-8B can answer correctly over 2 times in 4 trials based on $I', Q'$. In total, we collect 37k samples for training. In terms of data composition, approximately 60% of the source images come from SA-1B (Kirillov et al., 2023), while the remaining $\sim$40% are drawn from other public datasets (CC12M, LAION, MetaCLIP, Visual Genome, STPLS3D). To prevent data contamination, we apply strict image-hash deduplication against all evaluation benchmark test sets.

## B.2. Benchmark Construction

For benchmark construction, we adopt a more powerful MLLM, Gemini-2.5-Pro (Comanici et al., 2025), as both the question generator and answer generator using the *Region-to-Image Distillation* pipeline. Note that we split images into training and benchmark partitions, ensuring no overlap of images or QA pairs to avoid data leakage. To ensure benchmark's validity, we further perform human verification. Each example is independently checked by 3 paper authors (PhD-level researchers), who are provided with the full image, the corresponding cropped region, and the model-generated QA pair. Annotators **verify (not annotate)** that (i) the question is unambiguous and answerable, and (ii) the final answer is correct under both the full-image and cropped-region views. Each annotator spends only about 10 hours to verify roughly 650 raw samples (1,960 raw samples in total for 3 annotators). After careful verification and selection (filtering out very easy questions from the view of annotators), we finally remain 845 examples to form the *Zoom-Bench*, each consisting of a high-resolution full image and a small-ratio cropped view showing the key region for the question (i.e., the dual-view evaluation). Note that HR-Bench (Wang et al., 2025f), TextVQA-gt-bbox (Khayatkhoei et al., 2025), and TreeBench (Wang et al., 2025b) include evaluation settings that resemble our dual-view protocol. However, HR-Bench constructs its two views by cropping the same 8K image to 4K; since the crop covers a large portion of the original image (around 25% by area), the resulting global–local gap is typically small. TextVQA-gt-bbox and TreeBench (Wang et al., 2025b) provides a human-annotated ground-truth box for each question, which is costly to obtain at scale. Moreover, TreeBench (Wang et al., 2025b) mainly targets on the evaluation of "Thinking with Images" models and thus does not conduct the dual-view evaluation; HR-Bench (Wang et al., 2025f) and TextVQA-gt-bbox (Khayatkhoei et al., 2025) are relatively easy for current MLLMs (e.g., Qwen2.5-VL-7B, not the most advanced model, already achieves 84.9% on TextVQA-gt-bbox and 68.5% on HR-Bench-8K), making them less suitable for testing advanced and rapidly evolving models. We also demonstrate the data statistics of our benchmark in Figure 8. Besides, the benchmark comprises 224 open-ended questions with canonical target answers and 621 multiple-choice questions, covering 6 distinct dimensions.

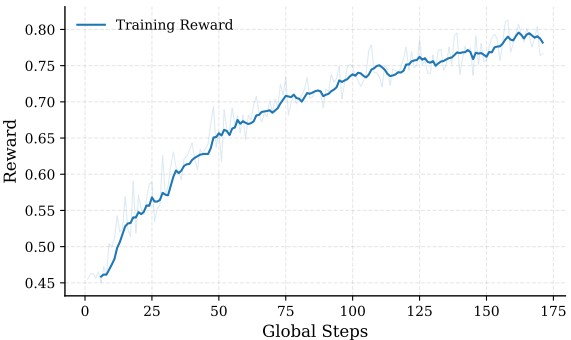

*Figure 9.* The training reward of Qwen3-VL-8B.

## C. Implementation Details of Experiments

### C.1. RL Training Settings

We adopt the EasyR1 (Zheng et al., 2025a) framework for multi-modal reinforcement learning, which is based on DAPO (Yu et al., 2025a). Specifically, we set the rollout group to 8, training epoch to 1, and use AdamW optimizer (Loshchilov & Hutter, 2019) with a learning rate of $1 \times 10^{-6}$, weight decay of $1 \times 10^{-2}$, rollout temperature of 1.0, and gradient clipping at a maximum norm of 1.0. Other hyperparameters follow the default settings provided in the EasyR1 framework. Besides, we all use the "Instruct" version of Qwen3-VL models with different sizes (4B, 8B, 235B) for training and evaluation.

### C.2. Hardware and Cost

Our training is conducted on 64×PPUs. RL training takes approximately 24 hours for ZwZ-8B on our dataset. On the data synthesis side, the entire pipeline costs only around $60 via API calls (∼2M calls in total), making it substantially cheaper than methods that require expensive human-annotated data (Zhang et al., 2025h). At deployment time, inference cost dominates, and our single-pass model is approximately 10× faster than agentic baselines that require repeated tool calls per query, yielding clear overall cost savings at scale.

### C.3. Reward Modeling in Reinforcement Learning and Benchmark Evaluation

To provide the verifiable reward for reinforcement learning (Guo et al., 2025) and benchmark evaluation, we implement a two-stage tiered reward system. This system prioritizes deterministic evaluation and falls back to heuristic or semantic methods only when necessary, ensuring a balance between accuracy and computational efficiency. The evaluation pipeline follows a three-step hierarchy:

**Rule-based Matching.** The system first attempts to evaluate the response using exact or symbolic matching via tools like `mathruler` (Zheng, 2025). If the extracted answer matches the ground truth exactly, a maximum reward (i.e., 1) is assigned.

**LLM-as-a-Judge.** If neither rules nor numerical parsing can resolve the correctness, we utilize an efficient LLM (e.g., Qwen3-8B (Yang et al., 2025a) for RL rewarding and Qwen3-30B (Yang et al., 2025a) for benchmark evaluation) as a judge to provide a binary 0/1 reward.

We also plot the training reward curve for Qwen3-VL-8B on our dataset in Figure 9 and observe a consistent, steady increase over time, indicating that the dataset is highly learnable for the model.

### C.4. Inference Protocol and Latency

For direct-answer models (including ours), inference consists of a single forward pass on the full image. Besides, for inference efficiency, we do not open the thinking mode for GLM-4.5V and Kimi-K2.5. For agentic baselines, we follow their official tool-use configurations (crop/zoom operations and the maximum number of steps) and report end-to-end wall-clock latency measured under the same hardware and batch size.

*Table 9.* Controlled experiment on TreeVGR-RL data to isolate the effect of bbox grounding. Training with vs. without bounding boxes yields negligible differences, confirming that our R2I gains primarily stem from zooming supervision rather than privileged bbox information.

| Setting | ZoomBench | HR-4K | CV-2D | VStar | Avg |
|---|---|---|---|---|---|
| Qwen3-VL-4B | 40.2 | 78.2 | 79.8 | 80.1 | 69.6 |
| + TreeVGR-RL w/ bbox | 44.7 | 79.1 | 80.8 | 84.8 | 72.4 |
| + TreeVGR-RL w/o bbox | 45.8 | 76.9 | 81.4 | 84.3 | 72.2 |

*Table 10.* Scalability of R2I across model sizes. Our method consistently improves performance from 2B to 32B parameters.

| Model | ZoomBench | HR-4K | VStar | CV-B. | Avg |
|---|---|---|---|---|---|
| Qwen3-VL-2B | 41.3 | 71.8 | 72.8 | 79.0 | 66.2 |
| **ZwZ-2B** | 53.5 | 77.0 | 82.7 | 83.4 | **74.1** |
| Qwen3-VL-32B | 49.3 | 82.0 | 88.5 | 86.8 | 76.7 |
| **ZwZ-32B** | 60.4 | 86.3 | 94.2 | 88.5 | **82.3** |

### C.5. Mechanism Attribution: Zooming Supervision vs. Bbox Grounding

A natural question is whether the performance gains of R2I stem from distilling zooming supervision or merely from the privileged bounding-box grounding provided during training. In our R2I pipeline, bbox grounding and zooming supervision are inherently coupled (removing bboxes would introduce referential ambiguity, confounding the comparison). Therefore, to isolate the contribution of bbox grounding alone, we design a controlled experiment using TreeVGR-RL (Wang et al., 2025b), an external dataset that independently provides exact bounding boxes for VQA tasks, without any zooming-based data synthesis.

Specifically, we train Qwen3-VL-4B on TreeVGR-RL data in two settings: (1) with bounding boxes overlaid on the image ("w/ bbox"), and (2) without any bounding boxes ("w/o bbox"). If bbox grounding were the primary driver of our gains, we would expect a substantial performance difference between these two settings.

As shown in Table 9, training with vs. without bboxes on TreeVGR-RL data yields negligible differences (Avg: 72.4 vs. 72.2), confirming that bbox grounding alone, without zooming-based data synthesis, provides no significant advantage. This establishes that the gains of our R2I method arise from distilling zoomed perceptual evidence during data synthesis, not from the privileged bbox information used during training.

### C.6. Scalability across Model Sizes

Our main experiments (Table 2) demonstrate consistent gains across 4B, 7B, and 8B model scales. To further verify the scalability of Region-to-Image Distillation, we conduct additional experiments on Qwen3-VL-2B and Qwen3-VL-32B, representing a much smaller and a much larger model, respectively.

As shown in Table 10, R2I improves Qwen3-VL-2B by +7.95 and Qwen3-VL-32B by +5.67 average points. Combined with our main 4B/7B/8B results, this confirms that the method generalizes well from 2B to 32B, with consistent gains observed across all tested scales.

## D. Prompts.

**Prompt of LLM-as-a-Judge.** The prompt used for LLM-as-a-Judge in RL reward modeling and benchmark evaluation is shown as follows.

Your task is to judge whether the response expresses the same meaning as the answer of a question.
The question is: {question}
The answer is: {gt}
The response is: {response}
Please check and compare them and then judge. Consider them equivalent if:
Different phrasings of the same answer (e.g., 5 people vs five people)
Slight variations in description that refer to the same entity
Minor differences in precision that don't change the core answer (such as two colors that are similar: e.g., brown and dark brown, white and off-white)
Do NOT consider them equivalent if:
They refer to different objects, numbers, or concepts
They represent different interpretations of the question.
If the response is correct, your output should be Yes. Otherwise, your output should be No.
Put your output within \boxed{}.

**Prompt of Question Generation.** We also present the prompt used for question generation in our *Region-to-Image Distillation*.

You are an expert specialist in generating Visual Question Answering (VQA) datasets. Your task is to generate three high-quality and valid questions based solely on the provided image. Reference Examples (use these for inspiration on questioning angles):
{examples_str}
**Core Generation Rules: 1. Image-based Questions:** All questions must be answerable by examining the image provided. The answer to each question must be identical, accurate, and concise. It can be a **short, factual, and concrete string** (e.g., a number, a noun, or text).
**2. Content Relevance:** Question types include, but are not limited to:
**Object Identification:** Identify the exact sub-component or item (e.g., 'What is the person holding?' Answer: 'Apple').
**OCR:** Recognizing text within the image.
**3. Quality Control:** If the image is of such low quality that it is impossible to generate meaningful questions, return an empty JSON list: `[]`.
**4. Diversity of Questions:** Aim for a diverse range of questions. This includes counting, spatial relationships, scene recognition, anomaly detection, shape, material, structure, etc.
**Output Format:**
Please carefully observe the image and generate your response in the following JSON format:

```
[
  {"question": "Question 1"},
  {"question": "Question 2"},
  {"question": "Question 3"}
]
```

# E. Case Study

We present some ambiguous cases and benchmark cases generated by MLLMs during our benchmark annotating process.

## E.1. Ambiguous Cases

Below we show two cases that are unambiguous in the cropped image but become ambiguous in the full image due to information loss from cropping. Therefore, we introduce explicit visual grounding by overlaying the target bounding box on the image to construct our training dataset.

**Ambiguous Case 1.**

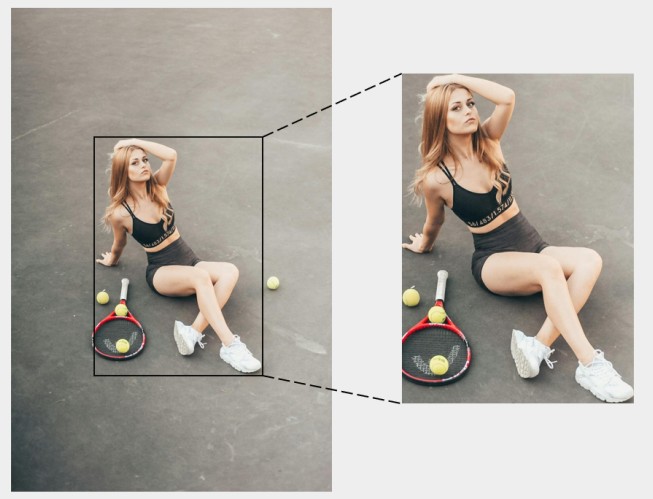

**Question:**
How many tennis balls are visible in the image?
A. 1 B. 2 C. 3 D. 4

**Answer:**
C for the cropped image and D for the full image

**Ambiguous Case 2.**

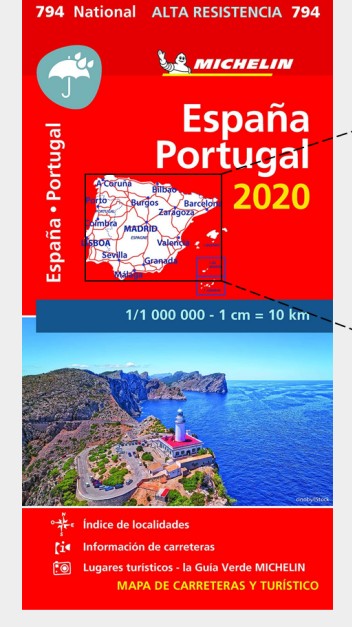

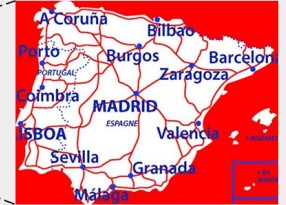

**Question:**
What is the name of the island group labeled in the bottom right corner of the map? (Answer the full text as written)
A. I. BALEARES B. I. DE AZORES C. I. DA MADEIRA D. I. DE LAS CANARIAS

**Answer:**
C for the cropped image. As for the full image, it should be "A CANARIAS".

**Ambiguous Case 3.**

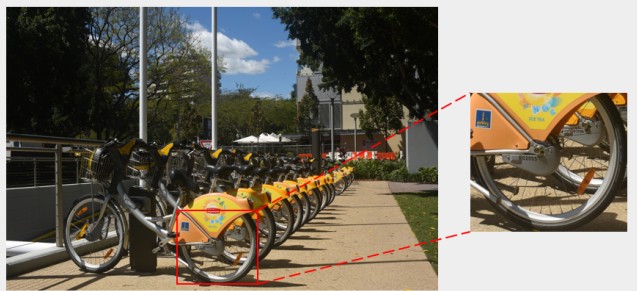

**Question:**
What brand name is written on the orange fender of the bicycle?
A. City Cycle B. Lipton C. ICE TEA D. Bike Share

**Answer:**
C for the cropped image and B for the full image.

## E.2. Benchmark Cases

We also provide one representative case for each category (including counting, OCR, color, structure, material, and identification) in our benchmark. Note that all questions are generated by MLLMs via our *Region-to-Image Distillation* method and are of high quality.

**Fine-Grained Counting.**

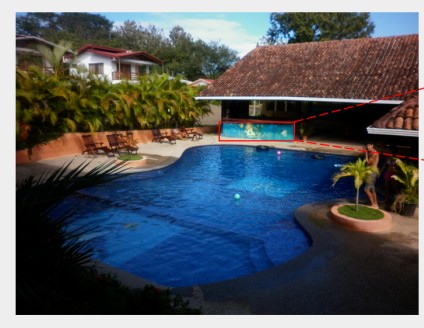
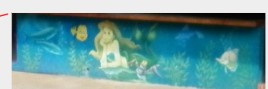

**Question:**
How many fish are visible on the left side of the mermaid?
A. 5 B. 4 C. 3 D. 2

**Answer:**
C

**OCR.**

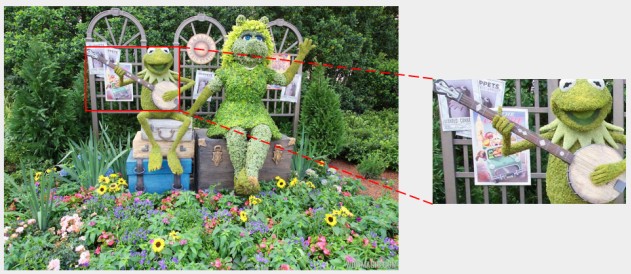

**Question:**
What word is visible on the poster behind the character that starts with 'WANTED'?
A. ROOST B. COST C. CONST D. GHOST

**Answer:**
C

**Color Attributes.**

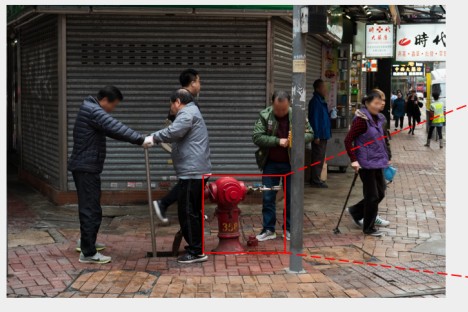 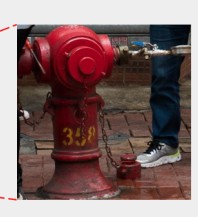

**Question:**
What is the color of the valve handle attached to the hydrant?
A. red B. blue C. yellow D. black

**Answer:**
B

**Structural Attributes.**

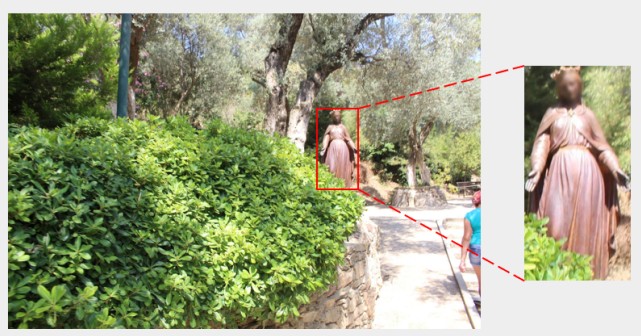

**Question:**
What type of headwear is on the statue?
A. wreath B. tiara C. headdress D. crown

**Answer:**
D

**Material Attributes.**

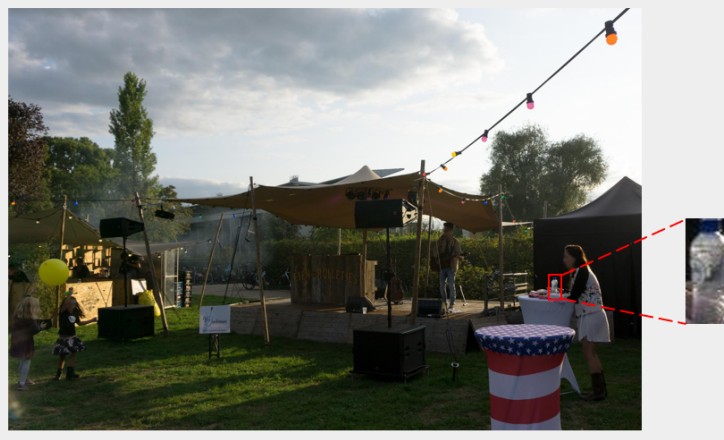

**Question:**
What is the primary material of the bottle shown in the image?
A. glass B. ceramic C. metal D. plastic

**Answer:**
D

---

**Object Identification.**

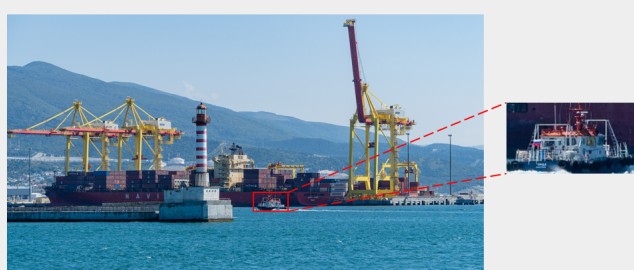

**Question:**
What flag is displayed on the stern of the boat?
A. Greek flag B. Turkish flag C. Russian flag
D. Ukrainian flag

**Answer:**
C

---

## F. Relative Attention Map Computation

We follow Khayatkhoei et al. (2025) to compute *relative attention* maps that highlight image regions that are specifically relevant to answering a given question, rather than reflecting question-agnostic or "baseline" attention patterns.

**Setup and notation.** Given an image-question pair $(x, q)$, an MLLM first encodes the image into a grid of $N \times N$ visual tokens (patch tokens) produced by the vision encoder (e.g., a ViT). Let $N^2$ denote the number of visual grid cells. Depending on the architecture, the ViT tokens are either (i) directly projected into the LLM input space (e.g., by an MLP), or (ii) resampled by a Transformer connector (e.g., a Q-Former) into $T$ *image tokens* that are then fed to the LLM together with the text tokens.

Let $L$ be the number of LLM layers and $H$ the number of attention heads per LLM layer. If a Transformer connector is present, let $L_c$ and $H_c$ be its number of layers and heads. We use greedy decoding and denote the special *starting answer token* position by $s$ (i.e., the first position at which the model starts generating the answer).

**(1) Answer-to-token attention in the LLM.** We extract the softmax cross-attention from the starting answer token to the $T$ image tokens in each LLM layer:

$$A_{\text{st}}(x, q) \in \mathbb{R}^{L \times H \times 1 \times T}. \tag{3}$$

We then average over heads to obtain

$$\hat{A}_{\text{st}}(x, q) = \frac{1}{H} \sum_{h=1}^{H} A_{\text{st}}^{(h)}(x, q) \in \mathbb{R}^{L \times 1 \times T}. \tag{4}$$

**(2) Token-to-image attention in the visual connector.** For models with a Transformer connector (e.g., Qwen-VL series), we extract the connector's softmax cross-attention from each image token to the $N^2$ ViT grid tokens:

$$A_{\text{ti}}(x) \in \mathbb{R}^{L_c \times H_c \times T \times N^2}. \tag{5}$$

We average over heads:

$$\hat{A}_{\text{ti}}(x) = \frac{1}{H_c} \sum_{h=1}^{H_c} A_{\text{ti}}^{(h)}(x) \in \mathbb{R}^{L_c \times T \times N^2}. \tag{6}$$

For architectures without a Transformer connector (e.g., when an MLP directly maps ViT tokens into LLM image tokens), we set $\hat{A}_{\text{ti}}(x)$ to the identity mapping from the $N^2$ visual grid tokens to themselves (i.e., $T = N^2$ and $\hat{A}_{\text{ti}}(x)$ is an identity matrix for each connector layer index).

**(3) Answer-to-image attention.** We combine the LLM answer-to-token attention and the connector token-to-image attention via a tensor product to obtain an *answer-to-image* attention map over the $N^2$ visual grid cells:

$$A_{\text{si}}(x, q) \in \mathbb{R}^{L \times L_c \times 1 \times N^2}, \qquad A_{\text{si}}^{(m,k)}(x, q) = \hat{A}_{\text{st}}^{(m)}(x, q) \, \hat{A}_{\text{ti}}^{(k)}(x), \tag{7}$$

where $m \in \{1, \ldots, L\}$ and $k \in \{1, \ldots, L_c\}$ index the LLM and connector layers, respectively, and the right-hand side is standard matrix multiplication over the shared dimension $T$. Finally, we reshape the last dimension $N^2$ back to an $N \times N$ grid to obtain a spatial map.

**(4) Relative attention.** Softmax attention may include components that are not specific to the question (e.g., global "register" tokens or generic image summarization). To emphasize question-relevant attention, we normalize $A_{\text{si}}(x, q)$ by the answer-to-image attention computed with a fixed generic instruction $q'$:

$$A_{\text{rel}}(x, q) = \frac{A_{\text{si}}(x, q)}{A_{\text{si}}(x, q')}, \tag{8}$$

where the division is element-wise. Following Khayatkhoei et al. (2025), we use a fixed $q'$ such as *"Write a general description of the image."* for all samples. In practice, to avoid numerical issues we compute

$$A_{\text{rel}}(x, q) = \frac{A_{\text{si}}(x, q)}{A_{\text{si}}(x, q') + \epsilon}, \tag{9}$$

with a small constant $\epsilon$ (e.g., $10^{-6}$).

**(5) Layer selection to obtain a single map.** The relative attention defined above yields a tensor $A_{\text{rel}}(x, q) \in \mathbb{R}^{L \times L_c \times 1 \times N^2}$. For downstream coverage computation, we convert it into a single $N \times N$ map by selecting a fixed LLM layer. Specifically, we use the relative attention from the 24-th LLM layer:

$$A_{\text{rel}}^*(x, q) = A_{\text{rel}}^{(m^*, k^*)}(x, q) \in \mathbb{R}^{N \times N}, \qquad m^* = 24, \tag{10}$$

where $k^*$ denotes the connector layer index (for architectures with a visual connector); if no connector is used, we set $L_c = 1$ and $k^* = 1$. We keep this layer choice fixed for all samples and all experiments.

### F.1. Demonstrations of Attention Map

We also provide qualitative attention-map comparisons between our ZwZ-8B and the base Qwen3-VL-8B. ZwZ-8B concentrates more relative attention on the annotated key region, indicating improved localization of task-relevant evidence.

**Object Identification.**

| Original Image | Qwen3-VL-8B | ZwZ-8B |

**Question:** What symbol is visible on the tram's front, to the left of the number 8275?

  **A.** European Union flag - blue background with yellow stars

  **B.** No Smoking symbol - red circle with diagonal line through cigarette

  **C.** International Symbol of Access (wheelchair symbol) - white wheelchair figure on blue background

  **D.** Emergency Exit symbol - green rectangle with white running figure

**Answer:** C

**Structural Attributes.**

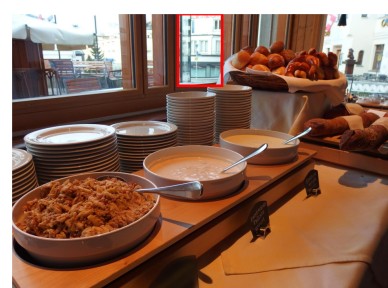 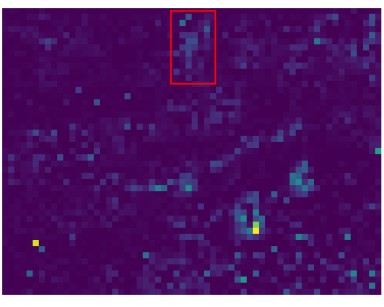 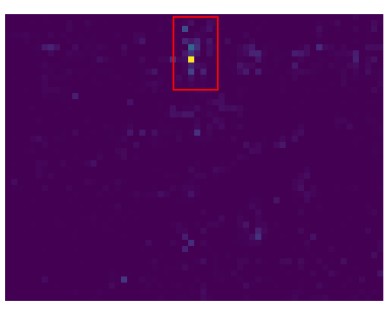

| **Original Image** | **Qwen3-VL-8B** | **ZwZ-8B** |
| --- | --- | --- |

**Question:** What is the shape of the blue sign on the pole?

 **A.** circle

 **B.** square

 **C.** inverted triangle

 **D.** rectangle

**Answer:** C

**OCR.**

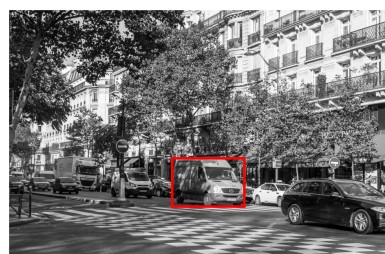 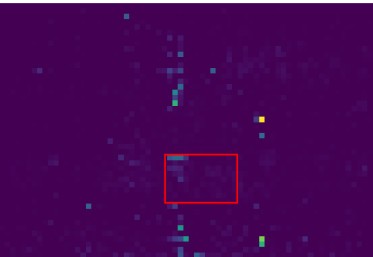 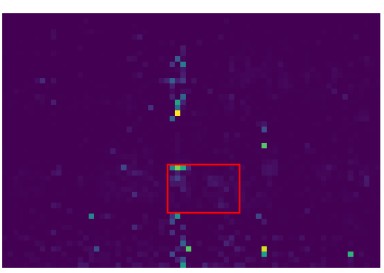

| **Original Image** | **Qwen3-VL-8B** | **ZwZ-8B** |
| --- | --- | --- |

**Question:** What is the text written on the side of the van, visible near the top?

 **A.** N° Indigo 0 820 20 15 55

 **B.** N° Indigo 0 820 20 15 50

 **C.** N° Indigo 0 820 20 14 40

 **D.** N° Indigo 0 820 20 16 60

**Answer:** B

