# OpenReview forum: "Zooming without Zooming: Region-to-Image Distillation for Fine-Grained Multimodal Perception"
_ICML.cc/2026/Conference — ICML 2026 regular_

### Official Review · Reviewer_FzcH · 2026-03-08

**Soundness:** 4
**Presentation:** 4
**Significance:** 3
**Originality:** 3
**Overall Recommendation:** 5
**Confidence:** 3

**Summary:**

This paper introduces a _Region-to-Image distillation_ method that distills the teacher’s region-level knowledge to the student model’s global-level views. The proposed method improves base models’ capabilities for fine-grained perception tasks and achieves better, or at least comparable, performances on various perception tasks, even compared to larger models.

**Compliance With Llm Reviewing Policy:**

Affirmed.

**Final Justification:**

All my concerns have been addressed. I'll keep my original positive rating score.

**Key Questions For Authors:**

1. In CV-Bench, there are tasks under the category of **Spatial Relationship** and **Relative Distance**. I’m interested in why the proposed _Region-to-Image distillation_ method would also improve the model's performance on such tasks where zooming into detailed regions does not seem to fully resolve the problem. Or is the performance improvement on this dataset attributed to the improvement in the tasks where zooming might be helpful, such as **Object Count** task?

2. For the **Regional-View** pipeline in _Section 5.1 Dual-View Evaluation_, did the model see only the cropped region, or did it still see the whole image with the bbox of the region?

3. During the training of ZwZ models, all synthetic training data was generated and labeled by the teacher model first. Could the authors provide further explanations why the ZwZ models achieve better scores on several general-purpose perception benchmarking datasets compared to their teacher models, Qwen3-VL-235B & GLM-4.5V (in Table 2, dataset **MicroP.** & **VStar** & **CV-B**)? And why does the ZwZ model sometimes perform better than Qwen3-VL-235B & GLM-4.5V in the regional-view in Table 6 (such as the category of **Color**)?

**Limitations:**

The authors didn't include a specific section on the limitations of the work, but I don't have anything to add either.

**Strengths And Weaknesses:**

Strengths:
- Through comprehensive, carefully designed experiments, the authors well justified the effectiveness of their proposed method.
- The proposed _Region-to-Image distillation_ method substantially improves the base models’ performance and even OOD generalizability.

Weaknesses:
- From my personal perspective, it’s not entirely fair to compare the inference time between models that require additional training and other training-free methods that only work at test time.

---

> ### Author Rebuttal · Authors · 2026-03-31
>
> **We thank the reviewer for the thoughtful and constructive feedback. We would like to address each concern below.**
>
>
> ---
>
> **W1: From my personal perspective, it’s not entirely fair to compare the inference time between models that require additional training and other training-free methods that only work at test time.**
>
> We would like to clarify that **our intended comparison is specifically against other training-based agentic models (e.g., DeepEyes, Thyme, SenseNova-MARS)**, since these methods share the same broad goal of improving fine-grained perception through additional agentic training. Training-free methods retain the clear advantage of requiring no extra training and occupy a different trade-off point entirely, so we do not position R2I as a direct competitor to them. Within the training-based regime, the core goal of R2I is to amortize repeated test-time zooming into a one-time offline training cost, so that the resulting model matches or exceeds agentic accuracy while running at base-model inference speed during deployment.
>
> ---
>
> **Q1: In CV-Bench, there are tasks under the category of Spatial Relationship and Relative Distance. I’m interested in why the proposed Region-to-Image distillation method would also improve the model's performance on such tasks. Or is the performance improvement on this dataset attributed to the improvement in the tasks where zooming might be helpful, such as Object Count task?**
>
> To better understand the source of improvement, **we have conducted a per-task breakdown on CV-Bench.** The results show that the gain is driven by two factors. **The dominant contribution indeed comes from counting tasks** (net advantage +39 questions), where R2I directly helps by providing clearer micro-evidence to distinguish and enumerate small or densely packed instances. **The gain also appears on spatial relationship tasks** (net advantage +5 questions). We attribute this to a more general effect of R2I: by training the model to recover task-relevant local evidence from the full image, R2I reduces interference from irrelevant global context and improves visual attention. For spatial reasoning, the key challenge is sometimes not the geometric comparison itself but correctly identifying which entities to compare, and R2I improves this object selection stage by allowing the model to focus on the relevant object pair before making the spatial judgment.
>
> ---
>
> **Q2: For the Regional-View pipeline in Section 5.1 Dual-View Evaluation, did the model see only the cropped region, or did it still see the whole image with the bbox of the region?**
>
> In the Regional View evaluation, **the model sees only the cropped region. It does not see the full image and it is not given a bounding box.** This design is intentional because the Regional View is meant to approximate an upper bound on the task once the decisive evidence has already been localized. In this setting, the model no longer needs to search for the relevant evidence and only needs to recognize or reason over it. The Global View evaluation then asks whether the same question can be answered from the original full image. The difference between the two reflects how much performance is lost due to the localization difficulty itself, which is exactly the bottleneck that R2I is designed to reduce.
>
> ---
>
> **Q3: Could the authors provide further explanations why the ZwZ models achieve better scores on several general-purpose perception benchmarking datasets compared to their teacher models, Qwen3-VL-235B & GLM-4.5V? And why does the ZwZ model sometimes perform better than Qwen3-VL-235B & GLM-4.5V in the regional-view in Table 6?**
>
> This is a key insight of our work. The teachers generate high-quality answers when given cropped micro-regions, a setting where fine details are unambiguous and hallucinations are minimized. However, the same teacher models perform significantly worse when given the full image, as evidenced by the large zooming gaps in Figure 5. **Our R2I distillation transfers the teachers’ regional-view expertise into the student’s global-view prediction ability, enabling the student to outperform the teachers under full-image inputs.**
>
> Furthermore, the ZwZ model’s ability to surpass its teachers, **even in the Regional View, can be attributed to the “Teacher Consensus” effect**. Our synthetic training data is not derived from a single source, but is cross-verified and labeled by a consensus of multiple frontier models (e.g., Qwen3-VL-235B and GLM-4.5V). This multi-teacher distillation acts as a denoising process, filtering out individual hallucinations or biases inherent in any single teacher. By learning from this refined, "voted" knowledge, the student can sometimes outperform an individual teacher even when evaluated on regional crops.

---

> > ### Author Rebuttal · Reviewer_FzcH · 2026-04-03
> >
> > Thanks for the response. All my concerns have been addressed. I'll keep my positive rating score.

---

> > > ### Author Response · Authors · 2026-04-03
> > >
> > > Thank you for your follow-up and for acknowledging our response. We're glad we were able to address your concerns. We sincerely appreciate the time and effort you've dedicated to this. Thanks again for your review and comments.

---

### Official Review · Reviewer_F1zJ · 2026-03-11

**Soundness:** 3
**Presentation:** 3
**Significance:** 3
**Originality:** 3
**Overall Recommendation:** 4
**Confidence:** 4

**Summary:**

The paper studies a practical bottleneck of current MLLMs: when the decisive evidence is tiny, single-pass inference from the full image often misses it, while “Thinking-with-Images” systems can zoom/crop iteratively but pay high latency due to repeated tool calls and re-encoding.
The main idea is to convert zooming into a training-time primitive via Region-to-Image Distillation (R2I): generate VQA questions/answers on micro-crops using strong teachers with consensus filtering, then “distill” them back to the full image by overlaying the target bounding box and adding a spatial constraint in the prompt.
The paper also introduces MicroPercept (845 examples) with a dual-view protocol (Global vs Regional) to quantify the “zooming gap”.
Experiments report consistent gains over Qwen-VL baselines and competitive or better results than some agentic zooming baselines, with lower inference latency.

**Compliance With Llm Reviewing Policy:**

Affirmed.

**Key Questions For Authors:**

1. Can the authors provide a clearer rationale for why bbox overlay (visual) outperforms bbox coordinates (text) beyond “strong privileged info”?

2. Can the authors provide some failure cases?

**Limitations:**

The main limitation lies in the efficiency claim, which is not strongly supported by the experimental results. That said, the analysis on bounding boxes, training data construction, and the attention map visualizations are quite interesting and provide useful insights. If the authors could clarify this point and provide additional supporting results, I would be happy to increase my score.

**Strengths And Weaknesses:**

Strengths
1. The training-time “zoom-in then distill back” framing is simple and intuitive.
2. MicroPercept is not large, but it appears to be reasonably well designed. The hybrid construction pipeline, the dual-view protocol, the coverage of six fine-grained dimensions, and the additional interpretability analysis based on relative attention coverage are all thoughtful and helpful.



Weaknesses
1. The paper motivates the work mainly from the efficiency angle, arguing that tool calling introduces substantial latency. However, this motivation is not fully reflected in the actual results. In Figure 1, the inference speed of the base model and the proposed method appears to be essentially the same.

2. In the experiments, the authors compare the proposed method with existing “Thinking-with-Images” agentic models. However, when the same MLLM backbone is used, the performance gap seems quite limited. For example, when both models are built on Qwen2.5-VL-7B, the proposed method and DeepEyes appear to achieve very similar results. It is also unclear which version of SenseNova-MARS is used; if it is also based on a Qwen3-VL backbone, then the performance may again be roughly on the same level.

3. In addition, the efficiency comparison in the figure only includes DeepEyes and Thyme, but does not include other relevant models such as mini-o3. This makes the efficiency comparison feel incomplete.

---

> ### Author Rebuttal · Authors · 2026-03-31
>
> **We thank the reviewer for the thoughtful and constructive feedback. We would like to address each concern below.**
>
> ---
>
> **W1: The efficiency motivation is not fully reflected — the proposed method and base model appear to have the same inference speed.**
>
> We would like to clarify that this is precisely our design goal. ZwZ retains almost the same single-pass inference speed as the base model, while significantly outperforming it in accuracy. **The efficiency gain is compared with "Thinking-with-Images" agentic models**, which require repeated tool calls and multi-turn visual re-encoding per query (~10× slower, as shown in Figure 1).
>
> ---
>
> **W2: Performance gap vs. agentic models with the same backbone seems limited.**
>
> We would like to clarify three points.
> - First, **SenseNova-MARS-8B is indeed trained from Qwen3-VL-8B backbone**.
> - Second, when comparing on the same backbone, **ZwZ achieves comparable or superior performance to agentic models while being ~10× faster**. Matching agentic accuracy at base-model speed is itself a meaningful result.
> - Third, and more importantly, there is a fundamental **data advantage** on the agentic models' side: models including DeepEyes, Thyme, SenseNova-MARS are all trained with large amounts of **expensive human-annotated or carefully curated data**, whereas ZwZ is trained exclusively on **fully synthetic data requiring zero human annotation**. Despite this disadvantage, ZwZ achieves competitive or superior results. Furthermore, our synthetic data is **complementary** to these curated datasets: mixing our R2I synthetic data with their human-annotated data would likely yield further gains.
>
> ---
>
> **W3: Efficiency comparison excludes models such as mini-o3.**
>
> Since these agentic baselines share similar multi-step tool-use inference patterns, we originally used two well-known models, DeepEyes-V2 and Thyme, as representative efficiency baselines. **We have now added latency measurements for two additional agentic models with strong performance, Mini-o3 and SenseNova-MARS-8B**, in **https://picui.ogmua.cn/s1/2026/03/30/69c9d9397239c.webp**
>
> The updated results lead to the same conclusion: our ZwZ models offer clearly better accuracy-speed trade-off while avoiding iterative tool calls.
>
> ---
>
> **Q1: Why does bbox overlay (visual) outperform bbox coordinates (text)?**
>
> We hypothesize this is because **when a bounding box is overlaid on the image, the vision encoder and cross modal projector can directly associate the marked region with the answer supervision**. By contrast, textual coordinates must first be parsed and then aligned back to visual tokens,  requiring the LLM to 'reason' about which visual tokens correspond to which numbers, which introduces an additional burden. In this sense, the bbox overlay on the image serves as a stronger signal during training, making it easier for the model to learn the correspondence between the global image and the relevant micro region.
>
> ---
>
> **Q2: Can the authors provide failure cases?**
>
> Yes. On MicroPercept (845 samples), ZwZ-8B shows 221 wins over Qwen3-VL-8B and only 56 losses, while 295 cases are failures shared by both models. This suggests that many remaining failures are not specific to ZwZ model, but stem from abilities that current MLLMs still generally struggle with.
>
> We also analyze some potential failure cases in our R2I method:
> - Global Relational Reasoning: While R2I excels at localizing and identifying micro-evidence, it can sometimes struggle with tasks requiring long-range relational reasoning between distant objects. The model’s enhanced focus on specific regions may not be enough to excellently solve such cases.
> - Teacher-Inherited Hallucinations: Since our student learns from teacher consensus, in rare cases where multiple teachers converge on the same incorrect micro-detail, the student may inherit this hallucination.
> - Fundamental Resolution Limits: When the original image is too high-resolution to be fed into the MLLM directly, it is typically downsampled, which can erase fine details in small regions.

---

> > ### Author Rebuttal · Reviewer_F1zJ · 2026-04-03
> >
> > Thanks to the authors for the detailed response. All my concerns have been addressed.

---

> > > ### Author Response · Authors · 2026-04-04
> > >
> > > Thank you very much for your thoughtful review and for the positive feedback on our rebuttal. We’re glad that our clarifications and additional results have addressed your concerns.
> > >
> > > We especially appreciate your openness to updating the score. If you feel appropriate, **we would be very grateful if you could adjust the score accordingly to reflect your current assessment.** Thank you again for your time and support!

---

### Official Review · Reviewer_q7e8 · 2026-03-13

**Soundness:** 2
**Presentation:** 4
**Significance:** 2
**Originality:** 1
**Overall Recommendation:** 3
**Confidence:** 4

**Summary:**

This paper studies fine-grained perception in multimodal large language models, where the decisive evidence is often local, small, and easily overwhelmed by global context. The paper proposes Region-to-Image Distillation (R2I), which converts inference-time zooming into a training-time primitive: a strong teacher answers questions on micro-cropped regions, and the resulting region-grounded supervision is distilled back to full-image inputs so that the student can improve fine-grained perception in a single forward pass. The paper also introduces a new benchmark and reports gains on multiple fine-grained perception tasks. Overall, I find the problem setting meaningful and the general direction interesting.

**Compliance With Llm Reviewing Policy:**

Affirmed.

**Final Justification:**

I thank the authors for their detailed rebuttal. After carefully considering all of their responses, I believe my current score remains reasonable, and I will keep it as my final score.

**Key Questions For Authors:**

- Can the authors provide a full cost ledger for the proposed pipeline, including teacher calls, total synthesis time, RL training GPU hours, and ideally a break-even analysis?
- Can the authors add a truly matched direct-synthesis baseline under the same teacher models, data scale, filtering rules, and training budget as the main R2I experiments?
- Can the authors provide stronger controls to separate the contribution of region supervision from that of privileged box grounding?
- Can the authors clarify which types of tasks can be amortized by this approach and which still fundamentally require test-time tool use?
- Can the authors provide a more complete dataset disclosure, including per-source composition, sampling ratios, deduplication / contamination controls, exact prompting/filtering/parsing templates, and licensing / redistribution status?
- In the *Zoom-in Synthesis on the Cropped Region* stage, how sensitive is the final performance to the sparsity threshold `τ`? A threshold sensitivity analysis would strengthen the paper.

**Limitations:**

yes

**Strengths And Weaknesses:**

**Strengths**
- The paper addresses a practically relevant problem: fine-grained evidence is often localized and difficult to capture in a single global pass.
- The proposed direction is reasonably novel. Instead of relying on iterative zooming at test time, the method amortizes part of this process into offline training through crop-based supervision and full-image distillation.
- The empirical coverage is fairly broad, including a self-constructed benchmark, external benchmarks, and several ablations.

**Weaknesses**
1. **The central efficiency narrative is not fully supported by the current evidence.** The paper mainly reports inference-time latency, while the proposed pipeline itself appears expensive due to strong teacher models, region proposal, multi-round answer generation, consensus filtering, and RL-style training. The contribution is therefore better characterized as *offline amortization for lower online latency*, rather than clearly lower overall cost.

2. **The paper does not yet establish novelty and superiority against the most relevant prior distillation-based alternatives.** Beyond the limited Direct Synthesis ablation, it would be important to position R2I against prior methods that also amortize expensive test-time procedures into training, such as Visual Program Distillation [1], as well as grounded-supervision or RoI-distillation approaches [2]. Without such comparisons or clearer discussion, it remains unclear how much of the contribution is methodologically new versus a task-specific instantiation of a broader distillation pattern.

3. **Part of the gain may come from privileged grounding rather than distilled zooming itself.** In particular, the stronger performance of `bbox-in-image` relative to `bbox-in-question` and `no-bbox` suggests that explicit spatial supervision may be an important factor, which weakens the claim that the main benefit comes specifically from internalizing zooming.

4. **The benchmark design is closely aligned with the method’s dual-view / zooming formulation.** Although the authors state that benchmark images do not overlap with training images and include human verification, the evaluation protocol may still favor the proposed inductive bias, making it harder to determine how well the method generalizes beyond this formulation.

5. **The disclosure of training-data details is still incomplete for a paper whose main contribution is a synthetic-data/distillation pipeline.** Key missing details include per-source composition and sampling ratios, deduplication and contamination-control procedures, and licensing / redistribution status.

---
**Reference**

[1] Visual Program Distillation: Distilling Tools and Programmatic Reasoning into Vision-Language Models, CVPR 2024 Oral

[2] GRIT: Teaching MLLMs to Think with Images, NeurIPS 2025

---

> ### Author Rebuttal · Authors · 2026-03-31
>
> **We thank the reviewer for the thoughtful and constructive feedback. We address each concern below.**
>
> ---
>
> **W1/Q1: Pipeline cost, full ledger.**
>
> We would like to clarify that R2I's overall cost is substantially lower than agentic alternatives.
> - Data Synthesis: **Unlike agentic methods requiring expensive human-annotated traces [1], R2I relies entirely on synthetic data**. Our dataset took ~2M API calls (just around $60) in two weeks: **a one-time expense.**
> - Training: Our single-pass model trains in 24 hours on 64 PPUs, **avoiding the slow multi-turn RL of agentic models.**
> - Inference: In deployment, inference costs dominate. Agentic models incur latency from repeated tool calls per query; **R2I only need a single forward pass (~10× faster). The one-time training cost becomes negligible over billions of inference queries.**
>
> [1] Thyme: Think beyond images. ICLR 2026
>
> **W2: Comparison with VPD and GRIT.**
>
> **VPD samples programs for existing human-curated VQA and distills verified execution traces** (i.e., programmatic reasoning). Crucially, its program filtering relies on human answers for correctness verification. **R2I is annotation-free**: both questions and answers are generated from zoomed-in regions, aiming to **distill local perception**. And our CoT comes from RL rollouts rather than programmatic traces.
>
> **GRIT** trains models to output reasoning chains with explicit bbox coordinates *at test time* for interpretable grounding. **R2I has a fundamentally different goal: it is a data synthesis method to internalize zooming benefits, also requiring no bbox coordinates generation at test time.**
>
> **Q2: Truly matched direct-synthesis baseline.**
>
> The "Direct Synthesis" ablation is precisely this baseline. The *only* difference is that it generates QA from the full image, whereas R2I generates QA from the micro-crop distilled back to the full image.
>
> **W3/Q3: Privilege information vs. distilled zooming.**
>
> In R2I, **bboxes solely resolve referential ambiguity** in our pipeline, a necessary step **strongly coupled with zooming supervision**. The “privileged information” view is introduced mainly to explain why adding bboxes during training does not cause adverse effects as no bboxes are provided at test time. **To fully isolate the effect of bbox as priviledge information, we train Qwen3-VL-4B with/without bboxes using TreeVGR-RL data [2] (providing exact bboxes for VQA)**. The negligible difference here confirms gains of R2I mainly stem from distilling zooming's perceptual details, not the priviledge information alone.
>
> |Model|MicroP.|HR-4K|CV-2D|Vstar|Avg|
> |-|-|-|-|-|-|
> |base|40.2|78.2|79.8|80.1|69.6|
> |w/ bbox|44.7|79.1|80.8|84.8|72.4|
> |w/o bbox|45.8|76.9|81.4|84.3|72.2|
>
> [2] Traceable Evidence Enhanced Visual Grounded Reasoning. ICLR 2026
>
> **W4: Benchmark design and inductive bias.**
>
> Our evaluation extends well beyond MicroPercept. **Our method consistently outperforms baselines on many external benchmarks (HR-Bench, Vstar, etc), confirming broad superiority and generalization.** Introducing a new diagnostic benchmark alongside diverse external evaluation is also standard practice [3,4].
>
> [3] Cambrian-1: A Fully Open, Vision-Centric Exploration of Multimodal LLMs. NeurIPS 2024
>
> [4] V*: Guided visual search as a core mechanism in multimodal llms. CVPR 2024
>
> **Q4: Amortized tasks vs. test-time tool use.**
>
> Tasks amenable to amortization are *information-neutral* (e.g., zooming, rotating): they reformat evidence already encoded in the input image. These transformations do not introduce new information, so their benefits can be internalized via R2I. Conversely, *information-gain* tools (e.g., web search) bring external new knowledge and thus still require test-time tool use.
>
> **W5/Q5: Data disclosure.**
>
> We will fully open-source our dataset. **Composition:** ~60% from SA-1B, ~40% from other public datasets (CC12M, LAION, etc.). **Contamination control:** strict image-hash deduplication against test sets. **Templates:** all prompts, filtering, and synthesis scripts are in our supplementary material. **License:** Apache-2.0 for research use.
>
> **Q6: Sensitivity to τ.**
>
> τ controls the maximum crop-to-image area ratio and **has a stable range**. However, excessively large τ introduces crops with large area ratios, yielding trivially easy or hallucinated questions that are filtered out or dilute training signal. Besides, extremely small τ leaves too few qualifying crops, causing most images to be discarded. Both cases reduce synthesis efficiency.
>
> We also test different τ. Using 10k samples from out dataset for τ=0.05 and τ=0.1, both give similarly strong results and **suggest that our default τ=0.1 is a suitable choice**, while τ=0.01 performs worse because only 584 samples satisfy this threshold in our dataset.
>
> |τ|MicroP.|HR-4K|CV-2D|Vstar|Avg|
> |-|-|-|-|-|-|
> |Qwen3-VL-4B|40.2|78.2|79.8|80.1|69.6|
> |0.01|43.4|80.0|81.0|86.4|72.7|
> |0.05|48.8|80.6|83.0|88.0|75.1|
> |0.10|49.7|80.6|82.1|88.0|75.1|

---

> > ### Author Rebuttal · Reviewer_q7e8 · 2026-04-04
> >
> > Thank you for the rebuttal; the added efficiency discussion, bbox-related controls, data disclosure, and threshold sensitivity analysis improve the paper’s clarity and address some of my secondary concerns. However, my main reservations remain unchanged: the efficiency narrative, the incremental novelty relative to related distillation/amortization-style methods, and the mechanism attribution behind the gains are still not sufficiently substantiated, so I will not raise my score.

---

> > > ### Author Response · Authors · 2026-04-04
> > >
> > > We sincerely thank the reviewer for the active discussion. We would like to address the remaining reservations.
> > >
> > > ---
> > >
> > > **1. Efficiency narrative.**
> > >
> > > The reviewer's initial concern is that *the pipeline appears expensive, and the contribution is better characterized as offline amortization rather than clearly lower overall cost.*
> > >
> > > We appreciate the reviewer’s perspective on offline amortization. We would like to further clarify that, as noted in our initial rebuttal to W1/Q1, **our method has a lower overall cost than agentic baselines across most stages**: data synthesis costs only ~$60 (vs. expensive human-annotated data), and both RL training and inference are more efficient due to the single forward pass (vs. repeated agentic tool calls).
> > >
> > > **Importantly, the “offline amortization” highlighted by the reviewer is indeed a key advantage of our method: at deployment scale, inference cost dominates, and our method achieves substantial savings in this stage. This can already lead to a clear overall cost benefit.** In addition, the one-time offline cost also remains low. We will clarify this efficiency discussion in the revised version.
> > >
> > > **2. Novelty relative to related distillation/amortization-style methods (VPD and GRIT).**
> > >
> > > We would like to further clarify the fundamental and significant differences in these methods:
> > >
> > > - **VPD distills high-level reasoning processes**: it teaches a model how to execute tool-chain programs by **distilling symbolic program execution traces from labeled VQA datasets** with ground-truth answer verification.
> > > - **GRIT trains a new output format**: it uses RL on **existing annotated VQA datasets** to teach models to **generate explicit bbox coordinates** interleaved with natural-language reasoning at test time.
> > >
> > > Unlike VPD and GRIT, our method does not distill a reasoning trace or train a new output format. Instead, **our method distills fine-grained local expertise (not the program execution traces) into the model's global perception, teaching it to recover fine-grained details from the full image without extra tool calls.** Our method also requires **no human annotations for training** and requires no spatial grounding tokens at test time.
> > >
> > > **3. Mechanism attribution behind the gains.**
> > >
> > > The reviewer's initial concern is that *the stronger performance of bbox-in-image relative to bbox-in-question and no-bbox suggests that explicit spatial supervision may be an important factor, weakening the claim that the main benefit comes from internalizing zooming.*
> > >
> > > As we clarified in our initial rebuttal, bboxes in our method solely resolve referential ambiguity (ensuring the model knows which object the region-level QA refers to). Thus, **explicit spatial supervision is strongly coupled with zooming supervision in our method**. The "privileged information" view is introduced to explain why adding bboxes during training does not cause adverse effects when no bboxes are provided at test time. Since bbox grounding and zooming supervision are inherently coupled in our pipeline (removing bboxes would introduce referential ambiguity, confounding the comparison), **we designed another controlled test to study the effect of explicit spatial supervision** using TreeVGR-RL [2], an external dataset that provides exact bboxes for VQA independently of zooming. This experiment (in our initial rebuttal to W3) directly validates our claim: **training with vs. without bboxes on external VQA data yields negligible differences (Avg: 72.4 vs. 72.3)**, confirming that bbox grounding alone, without zooming supervision, provides no significant gain. This establishes that **our gains arise from distilling zoomed perceptual evidence, not just from the bbox grounding.**
> > >
> > > As for why bbox-in-image outperforms bbox-in-question and no-bbox in our ablation (Table 5), we would like to further explain that this reflects how effectively the zooming supervision signal is delivered, not an independent source of gain. When a bbox is overlaid on the image, the vision encoder can directly associate the marked region with the answer supervision. By contrast, textual coordinates must first be parsed and aligned back to visual tokens, imposing an extra reasoning burden on the LLM. Thus, **bbox-in-image is simply a more efficient channel for delivering the same zooming supervision, not an independent "privileged grounding" advantage.**
> > >
> > > [1] Traceable Evidence Enhanced Visual Grounded Reasoning. ICLR 2026
> > >
> > > ---
> > >
> > > We sincerely hope that the evidence above substantively addresses the concerns. If the reviewer feels these remain insufficient, we would greatly appreciate more specific feedback on what additional analysis may be needed.

---

### Official Review · Reviewer_cezf · 2026-03-13

**Soundness:** 3
**Presentation:** 3
**Significance:** 3
**Originality:** 3
**Overall Recommendation:** 4
**Confidence:** 3

**Summary:**

The paper focuses on the problem of fine-grained perception and proposes Region-to-Image Distilation method that uses Object Segmentation model to crop a region and a strong teacher VLM to generate questions with answers residing in this small region. A student VLM is trained with the original images augmented with bounding boxes indicating regions of interest and a generated question. Results demonstrate that this approach results into stronger performance on several benchmarks compared to original base line models performance. Additionally, the paper contributes a manually reviewed MicroPercept benchmark.

**Compliance With Llm Reviewing Policy:**

Affirmed.

**Key Questions For Authors:**

Q1. How did you ensure that there was only one region that answers the question and it is the one that was cropped? Did you choose only the images where the main object is present once?

Q2. From the results it seems that on Specific Perception and OOD Generalization benchmarks higher performance is achieved by simply choosing larger model with more parameters. What are the gains of Region-to-Image Distilation for smaller/larger scale models?

Q3. How many GPUs of which kind did the training involve and for how long?

**Limitations:**

The limitations are not listed, however, while the framework brings an improvement for single region understanding, often crucial visual information resides in regions scattered across the image.

**Strengths And Weaknesses:**

S1. Well-written paper with logically chosen baselines and meaningfull experiments. Additional implementation details are well-documented in the appendix.

S2. The main concept of training the model with dedicated fine-grained perception data to achieve higher-performance in a single pass is impactfull

W1. It is not clear how the model deals with the ambiguous questions and how many of such are in the training dataset.

W2. To make a claim about generalizability of the proposed approach an analysis of how this approach affects performance of larger scale models would be required. It would be interesting to see how this training strategy affects larger scale models, especially considering their superior performance on the OOD and Specialized Benchmarks (it is understandable though that computational resources required for such training may not be available).

W3. The paper has no mention of hardware and training times involved into proposed training.

W4. Currently only works for single region crops, no experiments prooving otherwise found.

---

> ### Author Rebuttal · Authors · 2026-03-31
>
> **We thank the reviewer for the thoughtful and constructive feedback. We would like to address each concern below.**
>
> ---
>
> **W1/Q1: How did you ensure that there was only one region that answers the question, and how many ambiguous questions exist in the training dataset? Did you choose only the images where the main object is present once?**
>
> We address referential ambiguity through **two complementary mechanisms** as mentioned in Section 3.1.
>
> - First, during data synthesis, questions are generated exclusively from micro-cropped regions, ensuring **the question is strictly answerable from that specific crop alone**. Additionally, our multi-teacher consensus filtering (retaining only samples where >6/8 teacher responses agree) further eliminates ambiguous or invalid QA pairs before they enter training.
> - Second, we find that using MLLMs to correctly ensure that a main object is present only once is practically difficult. Thus, when distilling back to the full image, **we apply explicit visual grounding by overlaying the bounding box onto the full image and prepending a spatial constraint to the question prompt (e.g., "Only focus on the objects inside the bounding box")**. This grounding transformation resolves potential referential ambiguity that arises from zooming out.
>
>
> The combination of these mechanisms ensures that ambiguous questions are either resolved or discarded. We also provide concrete examples in Appendix D.1 to show how our approach resolves ambiguous cases by introducing explicit visual grounding on the image.
>
> ---
>
> **W2/Q2: What are the gains of Region-to-Image Distillation for smaller/larger scale models?**
>
> Our main paper already demonstrates consistent gains across three model scales (4/7/8B). To further address scalability concerns, we conducted additional experiments on **Qwen3-VL-2B and Qwen3-VL-32B**.
>
>
> |Model|MicroP.|HR-4K|VStar|CV-B.|**Avg**|
> |-|-|-|-|-|-|
> |Qwen3-VL-2B|41.3|71.8|72.8|79.0|**66.2**|
> |ZwZ-2B|53.5|77.0|82.7|83.4|**74.1**|
> |Qwen3-VL-32B|49.3|82.0|88.5|86.8|**76.7**|
> |ZwZ-32B|60.4|86.3|94.2|88.5|**82.3**|
>
> These results further confirm that R2I scales well: it improves Qwen3-VL-2B by +7.95 and Qwen3-VL-32B by +5.67 average points. Combined with our 4/7/8B results, **this shows that the method generalizes from 2B to 32B.**
>
> ---
>
> **W3/Q3: The paper has no mention of hardware and training times involved into proposed training.**
>
> Our training was conducted on 64×PPUs (The computing power of PPU is approximately 2/3 that of the A100). RL training takes approximately 24 hours for ZwZ-8B on our 37K-sample dataset.
>
> ---
>
> **W4/limitations: The framework currently only works for single region crops.**
>
> First, we would like to note that single-region fine-grained perception represents a dominant and practically critical failure mode for current MLLMs [1,2,3]. Improving this capability is therefore a highly impactful and non-trivial contribution in its own right.
>
> Second, and more importantly, **our single-region training can already generalize to multi-region perception scenarios**. As shown in Figure 4 and Table 2, ZwZ-8B consistently outperforms its base model on benchmarks that inherently require understanding spatially distributed information, including **GUI agent benchmarks** (e.g., ScreenSpot Pro, OSWorld) where relevant elements are scattered across the screen, **AIGC detection benchmarks** (e.g., FakeCLUE, LOKI-Image) requiring global-local consistency reasoning, aw well as **general multimodal understanding and reasoning benchmarks** (e.g., MMStar, BabyVision) involving diverse visual reasoning. These gains demonstrate that learning to reliably retrieve evidence from a single micro-region instills a broader perception ability, rather than overfitting to single-region patterns.
>
> Third, **our framework can be straightforwardly extended to multi-region training**: one can generate QA pairs that require jointly reasoning over two or more cropped regions (e.g., relational questions like "Is the object in region A larger than the object in region B?"), overlay multiple bounding boxes onto the full image during distillation, and train with such multi-region grounded supervision. We regard this as a promising and natural future direction.
>
> [1] Mmerealworld: Could your multimodal llm challenge highresolution real-world scenarios that are difficult for humans? ICLR 2025
>
> [2] Thyme: Think beyond images. ICLR 2026
>
> [3] Deepeyes: Incentivizing “thinking with images” via reinforcement learning. ICLR 2026

---

> > ### Author Rebuttal · Reviewer_cezf · 2026-04-03
> >
> > Thanks for providing detailed answers and additional experiment results! My questions are fully answered.

---

> > > ### Author Response · Authors · 2026-04-04
> > >
> > > Thank you for your positive feedback and for confirming that our rebuttal has addressed your concerns. We’re glad the additional clarifications and experiments were helpful.
> > >
> > > If you feel appropriate, **we would greatly appreciate it if you could adjust the score accordingly to reflect your current assessment.** Thank you again for your time and support!

---

### Decision · Program_Chairs · 2026-04-30

**Decision:**

Accept (regular)

**Comment:**

The paper proposes Zooming without Zooming, a Region-to-Image distillation method that effectively transfers region-level knowledge to global views, substantially improving base models' fine-grained perception and OOD generalizability. The submission receives overall positive ratings, with reviewers highly praising the comprehensive experiments and the impressive capability of the student models to match or outperform larger teacher models on various benchmarks. While the authors' rebuttal successfully clarifies secondary concerns regarding the efficiency narrative, bbox-related controls, and data disclosure, one reviewer retains reservations about the method's incremental novelty and mechanism attribution. Nevertheless, the AC concludes that the method's strong empirical effectiveness and robust performance gains significantly outweigh these limitations, and thus recommends acceptance.